# α-Synuclein fibrils subvert lysosome structure and function for the propagation of protein misfolding between cells through tunneling nanotubes

**Aysegul Dilsizoglu Senol[1]**, **Maura Samarani[1]**, **Sylvie Syan[1]**, **Carlos M. Guardia[2]**,
**Takashi Nonaka[3]**, **Nalan Liv[4]**, **Patricia Latour-Lambert[5]**, **Masato Hasegawa[3]**,
**Judith Klumperman[4]**, **Juan S. Bonifacino[2]**, **Chiara Zurzolo[1]**\*

**1** Unité de Trafic Membranaire et Pathogénèse, Département de Biologie Cellulaire et de l'Infection, Institut Pasteur, Paris, France, **2** Neurosciences and Cellular and Structural Division, Eunice Kennedy Shriver National Institute of Child Health and Human Development, National Institutes of Health, Bethesda, Maryland, United States of America, **3** Dementia Research Project, Tokyo Metropolitan Institute of Medical Science, Tokyo, Japan, **4** Section Cell Biology, Center for Molecular Medicine, University Medical Center Utrecht, Utrecht University, Utrecht, the Netherlands, **5** Dynamique des Interaction Hôte–Pathogène, Département de Biologie Cellulaire et de l'Infection, Institut Pasteur, Paris, France

☯ These authors contributed equally to this work.
\* chiara.zurzolo@pasteur.fr

**Data Availability Statement:** All relevant data are within the paper and its Supporting Information files.

## Abstract

The accumulation of α-synuclein (α-syn) aggregates in specific brain regions is a hallmark of synucleinopathies including Parkinson disease (PD). α-Syn aggregates propagate in a "prion-like" manner and can be transferred inside lysosomes to recipient cells through tunneling nanotubes (TNTs). However, how lysosomes participate in the spreading of α-syn aggregates is unclear. Here, by using super-resolution (SR) and electron microscopy (EM), we find that α-syn fibrils affect the morphology of lysosomes and impair their function in neuronal cells. In addition, we demonstrate that α-syn fibrils induce peripheral redistribution of lysosomes, likely mediated by transcription factor EB (TFEB), increasing the efficiency of α-syn fibrils' transfer to neighboring cells. We also show that lysosomal membrane permeabilization (LMP) allows the seeding of soluble α-syn in cells that have taken up α-syn fibrils from the culture medium, and, more importantly, in healthy cells in coculture, following lysosome-mediated transfer of the fibrils. Moreover, we demonstrate that seeding occurs mainly at lysosomes in both donor and acceptor cells, after uptake of α-syn fibrils from the medium and following their transfer, respectively. Finally, by using a heterotypic coculture system, we determine the origin and nature of the lysosomes transferred between cells, and we show that donor cells bearing α-syn fibrils transfer damaged lysosomes to acceptor cells, while also receiving healthy lysosomes from them. These findings thus contribute to the elucidation of the mechanism by which α-syn fibrils spread through TNTs, while also revealing the crucial role of lysosomes, working as a Trojan horse for both seeding and propagation of disease pathology.

**Funding:** Association pour la Recherche sur la Sclérose Latérale Amyotrophique et Autres Maladies du Motoneurone (ARSLA, appel d'offre 2016), INNOV 39-2019 Flash Start (Institut Pasteur) supported ADS, Access to Correlative Light and Electron Microscopy Flagship Node was supported by Euro-BioImaging grant (EuBI_AYDI109) for ADS, MS was supported by France Alzheimer (S-FB17026) and the Marie Skłodowska-Curie Action H2020-MSCA-IF-2019 EU project 897378, Agence Nationale de la Recherche (ANR-16-CE16-0019-01), Fondation pour la Recherche Médicale (FRM-2016-DEQ20160334896), LECMA-Vaincre Alzheimer (2016 / FR-16020) and France Alzheimer (AAP SM 2017#1674) foundations supported CZ, Intramural Program of NICHD (project ZIA HD001607) supported JSB, Grant-in-Aids for Scientific Research (JP26117005), CREST, JST (JP18071300) and AMED Brain/MINDS (JP18dm0207019) supported MH, Brain Science Foundation supported TN, NL was supported by the Netherlands Organization for Scientific Research (NWO) through a ZonMW-TOP grant (91216006) to JK. Agence Nationale de la Recherche (ANR-10-INSB-04), ANR/FBI and the Région Ile-de-France Domaine d'Intérêt Majeur-Malinf program supported the use of structured illumination microscopy (SIM) at the BioImagerie Photonic platform at Institut Pasteur. The funders had no role in study design, data collection and analysis, decision to publish, or preparation of the manuscript.

**Competing interests:** The authors declare no competing interests.

**Abbreviations:** α-syn, α-synuclein; AU, arbitrary unit; BSA, bovine serum albumin; CAD, Cath.a-differentiated; CathB, Cathepsin B; CLEM, correlative light-electron microscopy; DSHB, Developmental Studies Hybridoma Bank; EM, electron microscopy; ESCRT, endosomal sorting complexes required for transport; FBS, fetal bovine serum; FM, fluorescence microscopy; GA, glutaraldehyde; Gal3, Galectin-3; GFP, green fluorescent protein; KO, knockout; LAMP, lysosome-associated membrane protein; LC3, microtubule-associated protein 1A/1B-light chain 3; LLOMe, L-leucyl-L-leucine methyl ester; LMP, lysosomal membrane permeabilization; LysoTracker DR, LysoTracker Deep Red; ND, neurodegenerative disease; PB, phosphate buffer; PBS, phosphate buffered-saline; PCR, polymerase chain reaction; PD, Parkinson disease; PFA, paraformaldehyde; PM, plasma membrane; ROI, region of interest; ROS, reactive oxygen species; RT, room temperature; SB, Sleeping Beauty; SIM,

## Introduction

The accumulation of amyloidogenic proteins in different regions of the brain is a hallmark of neurodegenerative diseases (NDs) leading to cellular dysfunction, loss of synaptic communication, and deficits in specific brain functions [1,2]. The misfolded α-synuclein (α-syn) protein accumulates in intraneuronal inclusions known as Lewy bodies and Lewy neurites in NDs known as synucleinopathies, including the second most common ND, Parkinson disease (PD) [3–5]. These inclusions are associated with loss of neuronal and glial cells in specific brain regions, leading to both motor (e.g., bradykinesia, tremor, and rigidity) and nonmotor (e.g., cognitive impairment, depression, and anxiety) symptoms [6,7].

α-Syn is a small (140 amino acids) cytoplasmic protein enriched in the brain and localized to presynaptic terminals [8]. Although the precise function of α-syn remains poorly understood, it has been shown to promote membrane curvature, thus contributing to synaptic trafficking and vesicle budding [9–12], to modulate dopamine release through its association with presynaptic terminal SNARE complexes [13], and to be involved in synaptic attenuation [14]. A recent study has characterized in vitro a mechanism by which α-syn stabilizes, in a concentration-dependent manner, the docking of synaptic vesicles on the plasma membrane (PM) by establishing a dynamic link between the 2 membranes [15]. In PD brains, the spread of the pathology correlates with the presence of α-syn inclusions and follows a specific and predictable route through interconnected brain regions, mirroring the different stages of the disease [5,16]. These findings supported the hypothesis, later confirmed experimentally, that α-syn aggregates can be transferred from one cell to another both in vitro [17–22] and in vivo [23–27]. Furthermore, mounting evidence suggests that α-syn aggregates spread in a "prion-like" manner, as they are able to self-propagate by acting as "seeds" and inducing the misfolding of soluble native α-syn [19,27–35]. Thus, understanding how α-syn aggregates spread and how/where in the cell they induce misfolding of the soluble protein is crucial (i) to elucidate the pathogenesis of the disease; and (ii) to devise a therapy to inhibit the progression of the pathology. Different mechanisms depending on either cell-to-cell contact or secretion have been proposed for the spread of α-syn fibrils [35]. We and others have found that, in in vitro and ex vivo cultures, tunneling nanotubes (TNTs) play a major role in this process [19–21,36].

TNTs are actin-based, thin cellular protrusions that connect remote cells [37]. They have been identified in various cell types in vitro [38–41] and in vivo [42–47]. TNTs are open-ended structures [48,49] that allow the exchange of various cargos, including entire organelles, such as mitochondria and lysosomes, between cells [19,37–40]. In addition to α-syn fibrils, other amyloidogenic proteins such as prion protein [50–52], huntingtin [53], and tau [52,54,55] use TNTs as highways for transfer to naive cells, suggesting that TNTs are common routes for the spread of pathogenic proteins between cells. Of particular interest, we showed that α-syn fibrils can be efficiently transferred between neuronal Cath.a-differentiated (CAD) cells, primary neurons, primary astrocytes, and human neural progenitor cells and between organotypic hippocampal slices and astrocytes. We found that α-syn fibrils are mostly localized inside lysosomes in all of these cell types [19,21,56] and could be transferred inside lysosomes through TNTs from "infected" donor cells to naive acceptor cells in cocultures. Importantly, once arrived in the acceptor cells, α-syn fibrils were able to induce the aggregation of soluble α-syn [19]. How this occurs, however, is not clear.

Lysosomes are cytoplasmic, membrane-enclosed, acidic organelles that have long been considered as the "dustbin" and "recycling center" of the cells, as they are responsible for the degradation of both exogenous and intracellular cargos destined to lysosomes via endocytic, phagocytic [57,58], and autophagic pathways [59,60]. However, in recent years, mounting evidence revealed that lysosomes play a more complex role in the regulation of cellular

structured illumination microscopy; siRNA, small interfering RNA; sicontrol, scrambled siRNA; siTFEB, siRNA targeting TFEB; siArl8b, siRNA targeting Arl8b; SR, super-resolution; TBS, tris-buffered saline; TEM, transmission electron microscopy; TFEB, transcription factor EB; TNT, tunneling nanotube; WGA, wheat germ agglutinin; WT, wild-type.

homeostasis, as they are involved in crucial processes such as PM repair, transcriptional control, nutrient sensing, signaling, and energy metabolism [61–65]. It is therefore not surprising that lysosomal dysfunction is involved in a plethora of pathological conditions, including NDs [66]. Notably, the levels of lysosome-associated membrane proteins 1 and 2 (LAMP1 and LAMP2) and lysosomal enzymes Cathepsin B (CathB) and Cathepsin D (CathD) were found to be significantly reduced in PD patient brains, mouse, and cell models [67–72]. Moreover, lysosomal membrane permeabilization (LMP) has been reported in cells treated with protein aggregates including α-syn [73,74].

In the present study, we focused on the role of lysosomes in the intercellular spread of α-syn fibrils in neuronal cells. Particularly, we studied how α-syn fibrils affect lysosomes and how these organelles contribute to the spread and propagation of α-syn aggregates through TNTs. We found that α-syn fibrils alter the morphology of lysosomes and compromise lysosomal function. We demonstrated that α-syn fibrils induce the peripheral distribution of the lysosomes exhibiting functional impairment, likely mediated by transcription factor EB (TFEB), and showed that this peripheral positioning enhances the efficiency of α-syn fibrils' transfer in a cell-to-cell contact-dependent manner. In addition, we demonstrated that TFEB translocated to the nucleus in the presence of α-syn fibrils, probably contributing to the peripheral distribution of lysosomes. We also found that LMP, which is well described in cells following the uptake of α-syn fibrils from the medium, also occurs in acceptor cells following the transfer of lysosomes bearing α-syn fibrils from neighboring donor cells. Thus, by facilitating the escape of α-syn fibrils from lysosomes after their transfer, LMP appears to be a relevant mechanism for the seeding of soluble α-syn and propagation of the pathology in neighboring cells. Importantly, our super-resolution (SR) microscopy data suggest that lysosomes create an optimal environment for seeding and are a hub for the conversion of soluble α-syn into aggregates. Finally, by using a heterotypic cell coculture, we were able to track the origin and nature of the lysosomes transferred between cells. We found that cells overwhelmed by the presence of α-syn fibrils impairing lysosomal function transfer more damaged lysosomes to recipient cells, and, in return, they receive healthy lysosomes from neighboring cells devoid of α-syn, suggesting a potential rescue mechanism in α-syn pathology.

## Results

### α-Syn fibrils affect the morphology and function of lysosomes

We have previously reported that the majority of exogenous α-syn fibrils internalized by neuronal catecholaminergic cell line CAD localize to lysosomes [19]. We confirmed these findings by performing object-based 3D colocalization analysis (https://imaris.oxinst.com) in CAD cells challenged with α-syn fibrils for 18 hours prior to labeling for the lysosomal marker LAMP1. We detected 74 ± 1% of α-syn fibrils colocalizing with lysosomes (LAMP1+ organelles), confirming that the majority of α-syn fibrils localize to lysosomes following their uptake (Fig 1A). In addition, we found 60 ± 1% of lysosomes colocalizing with α-syn fibrils (Fig 1A). To further investigate the lysosomal localization of α-syn fibrils and the possible morphological consequences of this localization, we performed structured illumination microscopy (SIM). We found that α-syn fibrils localized either inside lysosomes (Fig 1B, magenta arrow and enlarged images on the right) or at the lysosomal membrane (Fig 1B, green arrow and enlarged images on the right). We then measured the diameter of lysosomes in control cells and in α-syn fibril-treated cells (where we distinguished between lysosomes containing or not containing α-syn fibrils). The average diameter of lysosomes in control cells and of lysosomes not containing α-syn fibrils in treated cells was almost identical (0.43 ± 0.01 μm), whereas lysosomes containing α-syn fibrils increased their diameter by 50% (0.63 ± 0.01 μm; Fig 1B, graph),

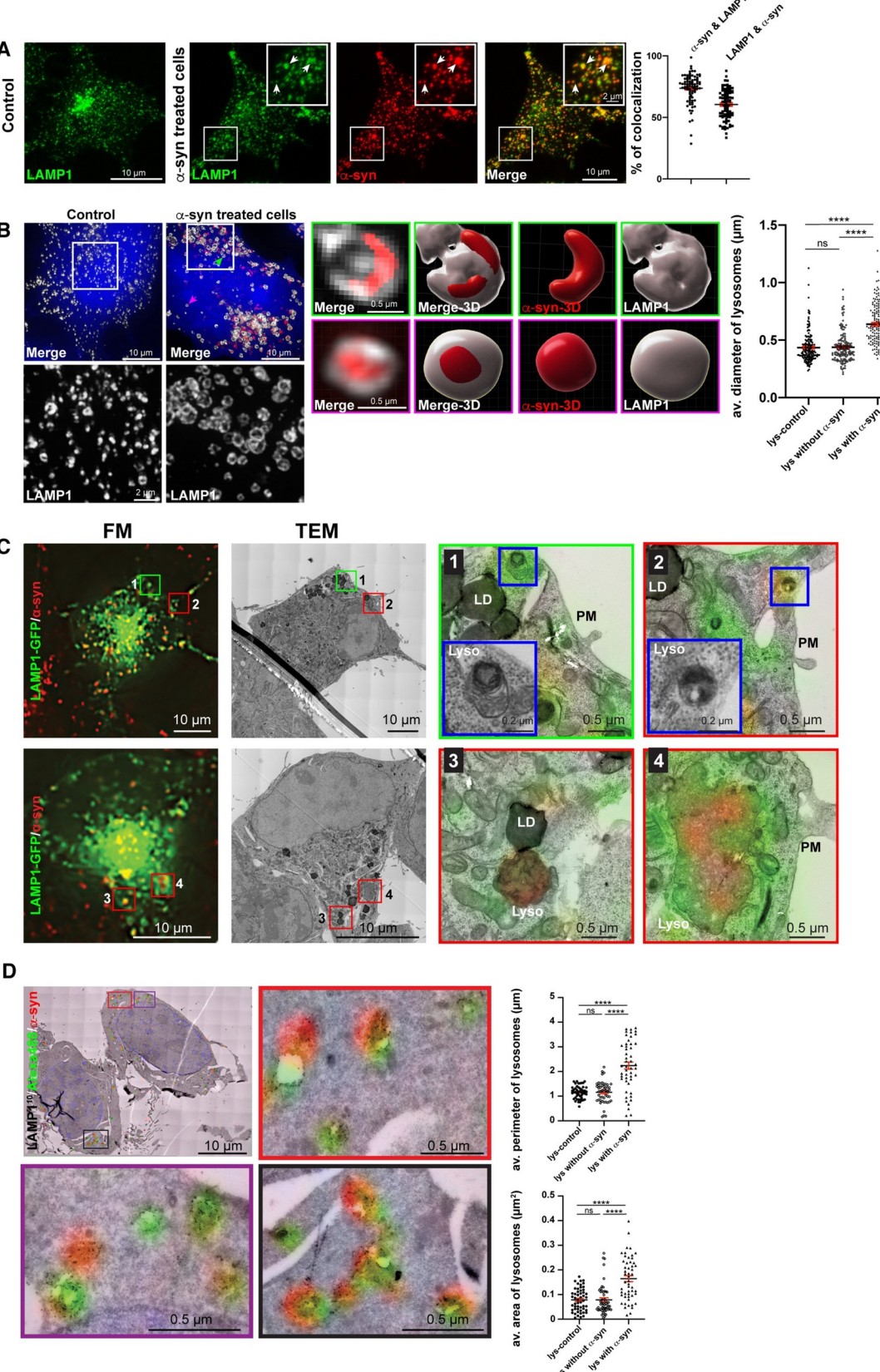

**Fig 1. α-Syn fibrils affect the morphology of lysosomes. (A)** Representative confocal images of control and Alexa 568–tagged α-syn fibril-treated (18 hours) CAD cells, immunolabeled for LAMP1 (green). Colocalization between LAMP1+ puncta and α-syn fibrils is indicated by arrows in the inset of a selected region delimited by a square. % of α-syn fibrils colocalizing with LAMP1+ puncta (74 ± 1%) and % of LAMP1+ puncta colocalizing with α-syn fibrils (60 ± 1%) performed by object-based 3D colocalization method (Imaris software) is presented. Mean ± SEM, $n$ = 3 (30 cells per condition). Scale bar: 10 μm (for inset: 2 μm). **(B)** SR images of control (left panel) and Alexa 568–tagged α-syn fibril-treated CAD cells for 18 hours (right panel), immunolabeled for LAMP1 (far-red) antibody (pseudo colored in gray). Merge images of each condition are presented with additional HCS CellMask Blue staining. Magenta and green arrows indicate the selected lysosomes having α-syn fibrils inside the lysosomal lumen and on the lysosomal membrane, respectively, where higher magnifications of these lysosomes and 3D reconstructions are represented in magenta and green squares, respectively. Scale bar: 10 μm (for insets: 2 μm, for magenta and green insets: 0.5 μm). Average diameter (μm) of lysosomes in control (0.43 ± 0.01) and in α-syn fibril-treated cells, the latter subgrouped as the following: Lysosomes without α-syn fibrils (0.43 ± 0.01) and lysosomes with α-syn fibrils (0.63 ± 0.01) are presented. Mean ± SEM, lysosomes' diameters were measured in SR images of 8 α-syn fibril-treated cells and 7 control cells (155 lysosomes per condition). Images were acquired by spinning disk microscopy with SR module. ns = not significant, ****$P$ < 0.0001 by Kruskal–Wallis nonparametric ANOVA test followed by Dunn multiple comparison tests. **(C)** FM image of LAMP1-GFP transfected and α-syn fibril-treated CAD cells for 18 hours (left panel). Enlarged merged images of correlative resin EM and FM images of 4 selected lysosomes not containing (number 1, indicated by green square) and containing α-syn fibrils (numbers 2, 3, and 4, indicated by red squares) are presented. Lysosomes number 1 and 2 are further magnified (blue squares) and presented within the same image. Scale bar: 10 μm (for red and green insets: 0.5 μm, for blue insets: 0.2 μm). **(D)** Overlay of on-section correlative EM images immunogold labeled for LAMP1[10] and FM images of LAMP1-GFP transfected and α-syn fibril-treated CAD cells (18 hours). Insets of 3 selected regions (indicated by red, purple, and black squares) are presented. Average perimeter (μm) of lysosomes in control (1.16 ± 0.04) and in α-syn fibril-treated cells: Lysosomes without α-syn fibrils (1.16 ± 0.06) and lysosomes with α-syn fibrils (2.24 ± 0.14; upper graph) are presented. ns = not significant, ****$P$ < 0.0001 by 1-way ANOVA followed by Tukey multiple comparison tests. Average area (μm$^2$) of lysosomes in control (0.08 ± 0.01) and in α-syn fibril-treated cells: Lysosomes without α-syn fibrils (0.08 ± 0.01) and lysosomes with α-syn fibrils (0.16 ± 0.01; lower graph) are presented. Mean ± SEM (50 cells per condition). ns = not significant, ****$P$ < 0.0001 by Kruskal–Wallis nonparametric ANOVA test followed by Dunn multiple comparison tests. Scale bar: 10 μm (for insets: 0.5 μm). The data underlying this figure may be found in S1 Data. α-syn, α-synuclein; CAD, Cath.a-differentiated; EM, electron microscopy; FM, fluorescence microscopy; LAMP, lysosome-associated membrane protein; LD, lipid droplet; Lyso and lys, lysosome; PM, plasma membrane; SR, super-resolution; TEM, transmission electron microscopy.

demonstrating that the enlargement induced by α-syn fibrils is restricted to the lysosomes containing fibrils.

To further investigate the morphology of compartments containing α-syn fibrils, we used a correlative light-electron microscopy (CLEM) approach to identify α-syn–positive and α-syn–negative lysosomes by fluorescence microscopy (FM) and study their corresponding morphology by electron microscopy (EM). First, we cultured CAD cells transiently transfected with a plasmid encoding LAMP1-GFP and loaded with Alexa 568–tagged α-syn fibrils on gridded coverslips and imaged α-syn fibril-treated cells in FM while marking coordinates of α-syn positive (Alexa568+/GFP+) and negative (Alexa568−/GFP+) LAMP1-GFP lysosomes. Following FM imaging, the CAD cells were prepared for EM by keeping their orientation, and serial sections were collected from the region containing the previously marked coordinates (see also Materials and methods) [75,76]. Then, the FM signal was correlated with EM images. Using this approach, we overcame the lack of suitable tools to identify α-syn fibrils directly by EM and were able to compare the structural properties of lysosomes containing or not containing α-syn fibrils. LAMP1-GFP–positive organelles colocalizing with α-syn fibrils in FM were structurally identified as lysosomes by the presence of electron-dense heterogenous luminal content, degraded membranes, and occasional intraluminal vesicles as morphological criteria [77]. Lysosomes not containing α-syn fibrils presented a healthy lysosome morphology with spherical shape and smooth membranes [77,78] (Fig 1C, green square, number 1). Interestingly, most lysosomes containing α-syn fibrils were found to be quite heterogeneous in structure (Fig 1C, red squares numbers 2, 3, and 4). Some had an unusual luminal content and displayed curvy limiting membrane, suggesting that the presence of the fibrils could deform the organelles (Fig 1C, red squares numbers 3, and 4). The aberrant, curvy organelles were completely absent in control cells not treated with α-syn fibrils (S1A Fig). A subset of

lysosomes containing α-syn fibrils, as defined by FM, had a healthy morphology (Fig 1C, red square, number 2), which could represent an early stage of fibrillar accumulation.

Next, we used another CLEM approach to analyze the morphology and size of α-syn–positive lysosomes with high resolution. CAD cells loaded with Alexa 568–tagged α-syn fibrils were prepared for EM following the flat-embedding Tokuyasu technique. This technique preserves the fluorescence signal of α-syn fibrils in ultrathin sections prepared for EM, and, in addition, allows to perform immunolabeling (both gold and fluorescent) of endogenous LAMP1 [79] (Fig 1D). Following a recently developed high-throughput on-section CLEM approach [80], we analyzed 50 LAMP1-immunolabeled lysosomes in control cells (S1B Fig) and 50 α-syn–positive and 50 α-syn–negative lysosomes in treated cells. For each lysosome, we correlated EM ultrastructural details (surface area and perimeter) and the presence of LAMP1 to the FM images showing the presence or absence of α-syn fibrils. Using this approach, we confirmed that the lysosomes bearing α-syn fibrils were significantly larger (perimeter: 2.24 ± 0.14 μm and area: 0.16 ± 0.01 μm$^2$) compared to lysosomes devoid of α-syn fibrils (perimeter: 1.16 ± 0.06 μm and area: 0.08 ± 0.01 μm$^2$) and control lysosomes (perimeter: 1.16 ± 0.04 μm and area: 0.08 ± 0.01 μm$^2$; Fig 1D, graphs).

In spite of these changes in lysosome morphology, the presence of α-syn fibrils had no effect on the abundance of lysosomes. Indeed, the average number of LAMP1-positive lysosomes per cell (control cells: 222 ± 6; α-syn fibril-treated cells: 218 ± 7; Fig 2A, left panel) and the level of LAMP1 (control: 0.9 ± 0.2 arbitrary unit [AU]; α-syn: 0.7 ± 0.1 AU; Fig 2A, right panel) did not show any significant difference in the presence or absence of α-syn fibrils. Overall, these results indicate that, while the overall amount of lysosomes is not affected by α-syn fibrils, lysosomes containing α-syn exhibit changes in size and morphology.

α-Syn fibrils have been shown to impair lysosomal function [67–72]. Thus, next we addressed their effect on lysosomes in our model system. We first assessed the acidity of the lysosomes in cells treated or not with α-syn fibrils by using a fluorogenic acidotropic dye, LysoTracker Deep Red (LysoTracker DR). The fluorescence intensity of LysoTracker DR was significantly decreased (approximately 41%) in α-syn fibril-treated cells compared to untreated control cells (Fig 2B), suggesting alkalinization of lysosomes in the presence of α-syn fibrils. Second, we used a DQ-BSA assay in which the fluorescent signal is detected only when the fluorogenic substrate is hydrolyzed by lysosomal proteases. We observed that the DQ-BSA fluorescence intensity was significantly decreased (approximately 45%) in the presence of α-syn fibrils compared to the control (Fig 2C), suggesting that the activity of the lysosomal proteases was compromised. Finally, we used a more specific approach to evaluate lysosomal function by quantifying the activity of the lysosomal enzyme CathB. The intracellular activity of CathB was detected as fluorescent puncta produced by the hydrolysis of the fluorogenic substrate Magic Red in live cells. We observed a significant decrease (approximately 42%) in CathB enzyme activity in cells challenged with α-syn fibrils compared to control cells (Fig 2D). Since the lysosomal pool was similar in control and α-syn fibril-treated cells, these data indicate that the decrease in the fluorescent signal of LysoTracker DR, DQ-BSA, and Magic Red is likely due to the presence of α-syn fibrils.

Altogether, our data indicate that the majority of the α-syn fibrils localize to the lysosomes, where they induce lysosomal enlargement and structural changes, and impair lysosome function.

## Lysosomes containing α-syn fibrils undergo lysosomal membrane permeabilization

α-Syn [73,74], as well as other amyloidogenic aggregates such as tau [81,82] and amyloid beta [83], induce LMP following their uptake from the culture medium in different cell types. This

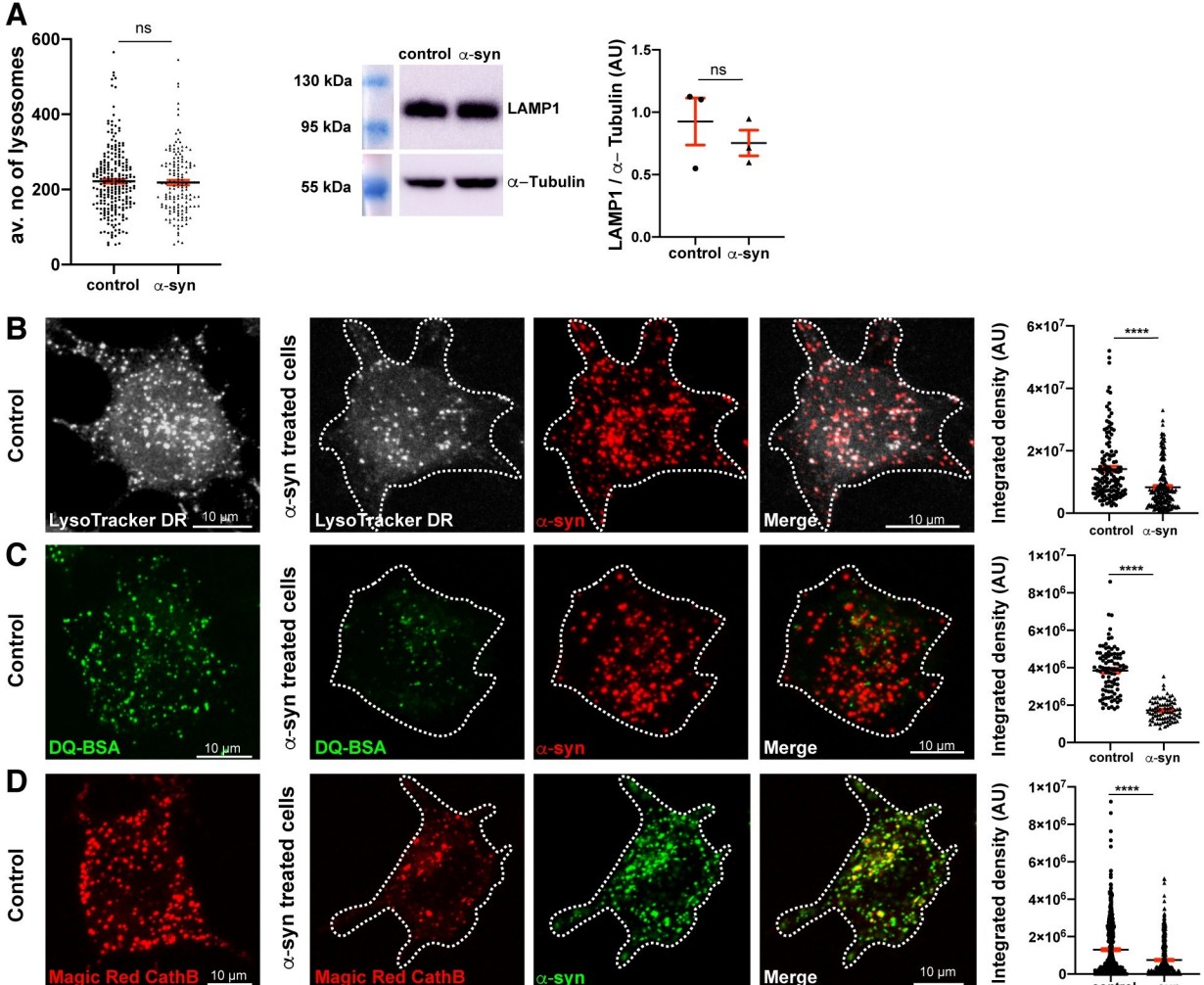

**Fig 2. α-Syn fibrils affect the function of lysosomes. (A)** Average number of lysosomes (LAMP1+ puncta) per cell in control (222 ± 6) and in α-syn fibril-treated CAD cells for 18 hours (218 ± 7) is presented. Mean ± SEM, $n = 5$ (50 cells per condition). ns = not significant by Whitney–Mann $U$ test (left panel). Western blot showing LAMP1 and α-tubulin expression in control and α-syn fibril-treated cells. Integrated density of LAMP1 protein expression levels normalized to α-tubulin in control (0.9 ± 0.2) and α-syn fibril-treated cells (0.7 ± 0.1) are presented in AU. Mean ± SEM, $n = 3$. ns = not significant by paired $t$ test. **(B)** Representative confocal images of control and Alexa 568–tagged α-syn fibrils challenged CAD cells (18 hours), labeled with LysoTracker DR (20 nM for 20 minutes; LysoTracker DR is pseudo colored in gray). Dashed lines represent the cell contour. Average integrated density of control cells ($1.4 \times 10^7 \pm 9 \times 10^5$) and α-syn fibril-treated cells ($8.3 \times 10^6 \pm 5 \times 10^5$) is presented. Mean ± SEM, $n = 3$ (40 cells per condition). $^{****}P < 0.0001$ by Mann–Whitney $U$ test. Scale bar: 10 μm. **(C)** Representative confocal images of control and Alexa 568–tagged α-syn fibrils (red) challenged CAD cells for 18 hours, treated with DQ-BSA (green) for 90 minutes. Dashed lines represent the cell contour. Average integrated density measured for control cells ($3.8 \times 10^6 \pm 1 \times 10^5$) and α-syn fibril-treated cells ($1.7 \times 10^6 \pm 6 \times 10^4$) is presented. Mean ± SEM, $n = 3$ (25 cells per condition). $^{****}P < 0.0001$ by Mann–Whitney $U$ test. Scale bar: 10 μm. **(D)** Representative confocal images of control and Alexa 488-tagged α-syn fibrils (green) challenged CAD cells for 18 hours treated with 1X Magic Red CathB (red). Dashed lines represent the cell contour. Average integrated density measured for control cells ($1.3 \times 10^6 \pm 6 \times 10^4$) and α-syn fibril-treated cells ($7.5 \times 10^5 \pm 5 \times 10^4$) is presented. Mean ± SEM, $n = 5$ (100 cells per condition). $^{****}P < 0.0001$ by Mann–Whitney $U$ test. Scale bar: 10 μm. The data underlying this figure may be found in S1 Data. α-syn, α-synuclein; AU, arbitrary unit; CAD, Cath.a-differentiated; CathB, Cathepsin B; LAMP, lysosome-associated membrane protein; LysoTracker DR, LysoTracker Deep Red.

allows escape of α-syn aggregates from the lysosome to act on the soluble α-syn in the cytosol. However, the fate of these lysosomes has not been assessed. Furthermore, we previously demonstrated that α-syn spread between cells occurs mainly by direct cell-to-cell transfer of lysosomes bearing aggregates through TNTs, cytoplasmic protrusions extending from the cell periphery and connecting distant cells [21,35,52] (S2 Fig). Therefore, the question arises as to

whether LMP also occurs after lysosomes bearing α-syn have been transferred to healthy neighboring cells. To address this question, we assessed first whether α-syn fibrils also induced LMP in our neuronal CAD cells. We used the Galectin-3 (Gal3) puncta assay based on the localization of Gal3 to damaged lysosomes. CAD cells were transiently transfected with Gal3-GFP and challenged or not with α-syn fibrils for 18 hours, prior to labeling for LAMP1 (Fig 3A, S3A Fig, number 1). After image acquisition by confocal microscopy, we quantified Gal3-GFP puncta formation in α-syn fibril-treated and untreated cells. We detected a significantly higher percentage of cells exhibiting Gal3-GFP puncta in the presence of α-syn fibrils (46 ± 2%) compared to the control condition (7 ± 1%) (Fig 3B, left graph), in which most cells displayed cytosolic Gal3-GFP [73,74]. However, in cells exhibiting Gal3-GFP puncta, there was no significant difference in the average number of puncta between control and α-syn fibril-treated cells (8 ± 1 and 12 ± 1, respectively; Fig 3B, right graph). In addition, we performed object-based 3D colocalization analysis to evaluate the association of Gal3-GFP with both α-syn fibrils and LAMP1+ lysosomes. We detected 38 ± 4% of Gal3-GFP puncta colocalizing with α-syn fibrils, 47 ± 3% of Gal3-GFP puncta colocalizing with LAMP1+ lysosomes, and 35 ± 3% of Gal3-GFP puncta colocalizing with both α-syn fibrils and lysosomes (Fig 3A, magenta arrowheads, and 3C). These findings demonstrate that more than one-third of Gal3-GFP puncta are recruited to the ruptured lysosomes containing α-syn fibrils. Furthermore, by using SIM, we could clearly observe the recruitment of Gal3-GFP to the damaged lysosomes positive for α-syn fibrils, which appeared to be either inside their lumen (Fig 3D) or at their membrane (Fig 3E).

Since lysosomes under LMP can be targeted to the autophagy pathway in order to be degraded (a process called lysophagy) [84], next, we investigated this possible scenario in CAD cells treated with α-syn fibrils in comparison with control cells. As a positive control, we treated CAD cells with L-leucyl-L-leucine methyl ester (LLOMe), which is a well-known inducer of LMP. Cells were then labeled for ubiquitin, microtubule-associated protein 1A/1B-light chain 3 (LC3), and LAMP1, and colocalization of these markers were analyzed. As expected, in CAD cells treated with LLOMe, we could detect good colocalization between LC3, ubiquitin and LAMP1, whereas in control cells, we could only detect a few LC3 and ubiquitin puncta and no colocalization between these puncta and lysosomes (S4 Fig). In 3 independent experiments, we could only detect 1 lysosome that colocalized with LC3 and ubiquitin in α-syn fibril-treated cells (S4 Fig); thus, lysophagy was an extremely rare event under these conditions and was not statistically different from the control condition.

## α-Syn fibrils seed de novo formation of aggregates at the lysosomes

LMP has been proposed to be a relevant mechanism for α-syn fibrils to escape from lysosomes and interact with soluble α-syn to induce seeding [19,85]. Therefore, we investigated the seeding activity of exogenous α-syn fibrils in CAD cells. To this aim, CAD cells were transiently transfected with a plasmid encoding α-syn-GFP, a soluble fluorescent reporter of α-syn, and the cells were challenged or not with Alexa 568–tagged α-syn fibrils for 18 hours. The formation of α-syn-GFP aggregates detected as green puncta was then quantified (Fig 4A, S3A Fig, number 2). We found that 76 ± 5% of cells treated with α-syn fibrils produced newly formed α-syn-GFP aggregates, whereas this percentage was significantly lower (13 ± 2%) in control cells transfected with α-syn-GFP only (Fig 4B, left graph). In addition, the average number of newly formed α-syn-GFP puncta detected in the cells treated with the fibrils (27 ± 3) was significantly higher than the α-syn-GFP puncta detected in the control condition (12 ± 1; Fig 4B, right graph), suggesting that the exogenous α-syn fibrils induce efficient seeding of the soluble protein. These data were in line with previous findings [86–89]; however, the question remains

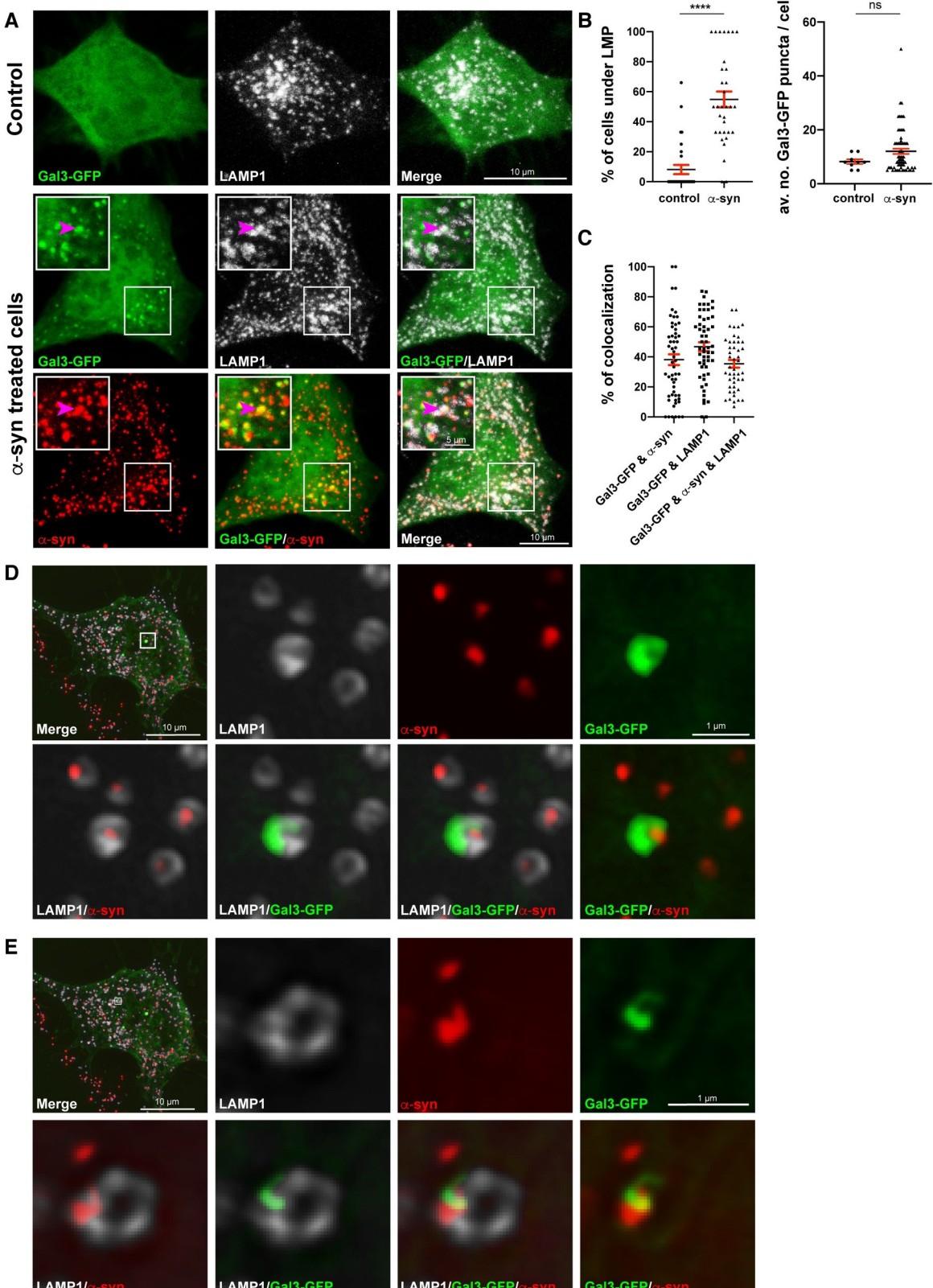

**Fig 3. α-Syn fibrils induce LMP. (A)** Representative confocal images of Gal3-GFP transfected control (upper panel) and Alexa 568–tagged α-syn fibril-treated CAD cells for 18 hours (lower panel), labeled with LAMP1-Alexa 647 antibody (pseudo colored in gray).

Higher magnifications of the selected region in α-syn fibril-treated cells (white squares) are presented on the upper left part of each image. Arrowheads indicate a lysosome having α-syn fibrils colocalizing with Gal3 puncta. Scale bar: 10 μm (for insets: 5 μm). **(B)** % of control (7 ± 1%) and α-syn fibril-treated CAD cells for 18 hours (46 ± 2%) under LMP (left graph), and average number of Gal3-GFP puncta for control (8 ± 1) and α-syn fibril-treated cells (12 ± 1) under LMP (right graph). Mean ± SEM, $n$ = 3 (50 cells per condition). ****$P$ < 0.0001 by Mann–Whitney $U$ test. **(C)** % of Gal3-GFP puncta colocalizing with α-syn fibrils (38 ± 4%), with LAMP1 (47 ± 3%) and with both α-syn fibrils and LAMP1 (35 ± 3%) calculated by object-based 3D colocalization (Imaris software). Mean ± SEM, $n$ = 3 (30 cells per condition). **(D)** SR image showing the recruitment of Gal3-GFP to a lysosome containing α-syn fibrils puncta in its lumen. **(E)** SR image showing the recruitment of Gal3-GFP to a lysosome having α-syn fibrils at its membrane. In D and E, insets of 2 selected regions (white squares) from different Z-planes of the same Gal3-GFP transfected and α-syn fibril-treated CAD cell, labeled for HCS CellMask Blue, DAPI, and LAMP1 (pseudo colored in gray) are presented. Images were acquired with SR SIM (Zeiss, LSM 780-Elyra PS.1). Scale bars: 10 μm (for insets: 1 μm). The data underlying this figure may be found in S1 Data. α-syn, α-synuclein; Gal3, Galectin-3; CAD, Cath.a-differentiated; LAMP, lysosome-associated membrane protein; LMP, lysosomal membrane permeabilization; SIM, structured illumination microscopy; SR, super-resolution.

as to whether lysosomes participate to this seeding event and how/where in the cell it occurs. To investigate whether seeding is associated with the lysosomes, we performed SR microscopy in CAD cells transfected with α-syn-GFP, loaded with α-syn fibrils and labeled for LAMP1. We observed 3 different scenarios: (i) lysosomes with both α-syn fibrils and α-syn-GFP aggregates colocalizing at the lysosomal membrane (Fig 4C); (ii) lysosomes with α-syn fibrils and α-syn-GFP aggregates colocalizing inside their lumen (Fig 4D); and (iii) lysosomes containing both α-syn fibrils and α-syn-GFP aggregates inside their lumen, but not necessarily colocalizing (Fig 4E). In addition, we performed object-based 3D colocalization analysis between newly formed α-syn aggregates (α-syn-GFP puncta), α-syn fibrils, and lysosomes. We observed that 69 ± 6% of the newly formed aggregates colocalized with LAMP1+ lysosomes, 74 ± 7% colocalized with α-syn fibrils, and 63 ± 6% of them were positive for both α-syn fibrils and LAMP1 + organelles (Fig 4F), indicating that in CAD cells, the majority of the seeding takes place at lysosomal compartments following α-syn fibrils uptake.

To understand whether the localization of α-syn-GFP aggregates to lysosomes was consistent with the beginning of the seeding event and not a post-seeding delivery of newly formed fibrils, we monitored the occurrence of seeding by live imaging microscopy. We first transfected CAD cells with soluble α-syn-GFP, and then, immediately after the addition of α-syn fibrils, we started to monitor the seeding event by detecting the formation of green puncta (α-syn-GFP newly formed aggregates) in these cells compared to control cells by live spinning disk microscopy. We found that the seeding first became apparent about 2 hours from the addition of the α-syn fibrils and increased with time (S1 Movie, S5A Fig). We did not observe any newly formed α-syn-GFP puncta in untreated control cells (S2 Movie, S5B Fig). Next, we performed SR microscopy in order to investigate whether these newly formed aggregates were associated with the lysosomes by immunolabeling LAMP1 at different time points (1 hour, 2 hours, and 4 hours). Interestingly, at all these time points (even in the rare events found at 1 hour), we found that newly formed aggregates colocalized with α-syn fibrils and LAMP1 (Fig 4G–4I). Therefore, newly formed α-syn-GFP aggregates following α-syn fibrils addition are associated with lysosomes at the very beginning of the seeding event, supporting our hypothesis that lysosomes function as hubs for the seeding of newly formed α-syn aggregates.

Overall, these results indicate that exogenous α-syn fibrils induce de novo aggregation of soluble α-syn in association with lysosomes.

## α-Syn–loaded lysosomes localize to the cell periphery and are less functional

The results presented above highlight the role of lysosomes in the propagation of α-syn misfolding after uptake of the fibrils from the culture medium. However, they do not address (i)

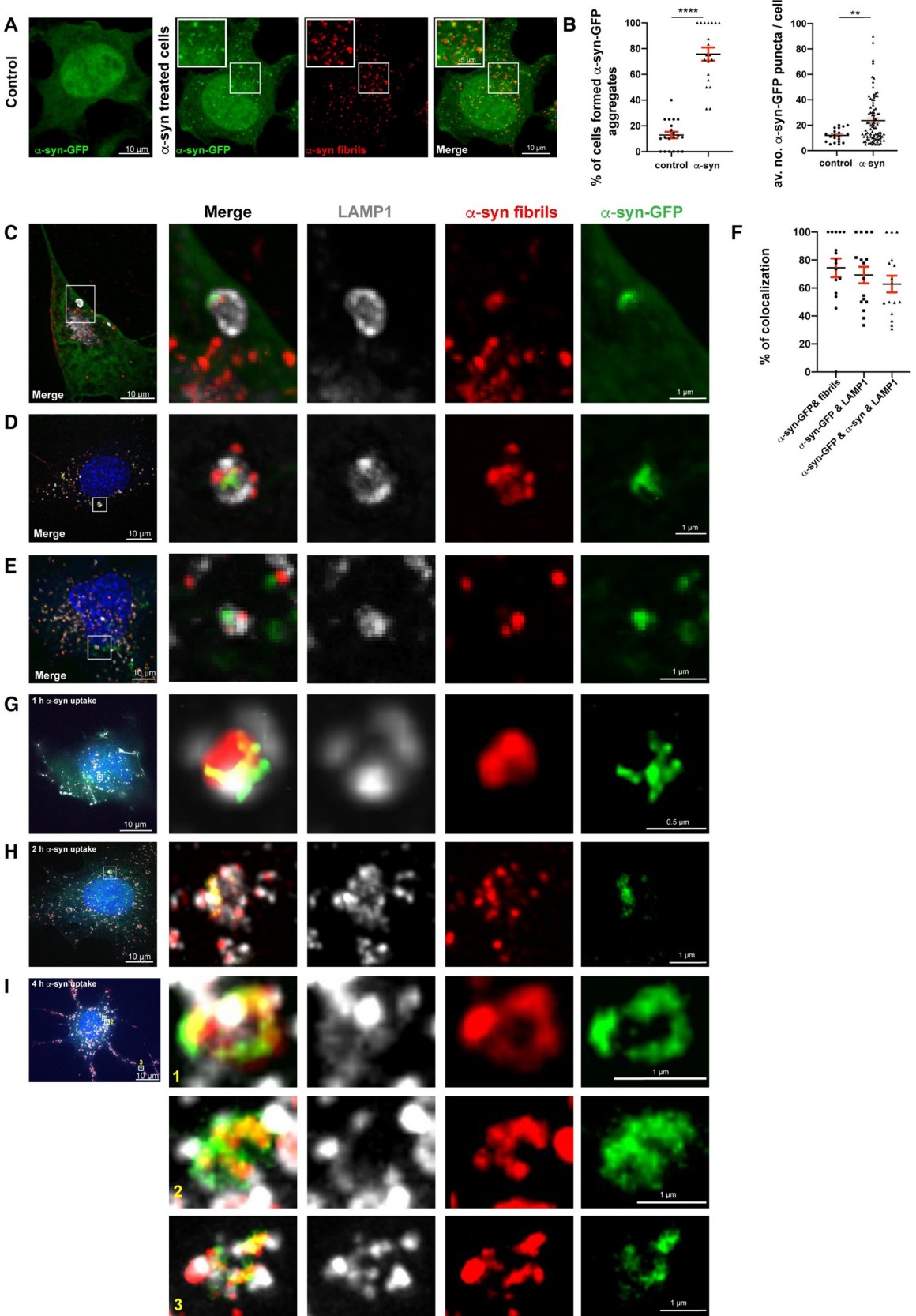

**Fig 4. α-Syn fibrils seed de novo formation of aggregates associated with lysosomes. (A)** Representative confocal images of α-syn-GFP transfected control or Alexa 568–tagged α-syn fibril-treated CAD cells (18 hours). Scale bar: 10 μm (for insets: 5 μm). **(B)** % of control (13 ± 2%) and α-syn fibril-treated CAD cells for 18 hours (76 ± 5%) that formed α-syn-GFP aggregates (left graph). Mean ± SEM, $n = 3$ (45 cells per condition). ****$P < 0.0001$ by 2-tailed $t$ test. Average number of α-syn-GFP puncta per cell in control (12 ± 1) and α-syn fibril-treated cells (27 ± 3) (right graph). Mean ± SEM, $n = 3$ (45 cells per condition). **$P = 0.005$ by Mann–Whitney $U$ test. **(C)** SR image showing α-syn fibrils puncta and α-syn-GFP+ aggregate colocalizing on the lysosomal membrane in CAD cells treated with α-syn fibrils for 18 hours. **(D)** SR image showing colocalization between α-syn fibrils puncta and α-syn-GFP+ aggregate inside the lysosomal lumen and α-syn fibrils on the lysosomal membrane in CAD cells treated with α-syn fibrils for 18 hours. **(E)** SR image showing α-syn fibrils puncta and α-syn-GFP+ aggregate inside the lysosomal lumen in CAD cells treated with α-syn fibrils for 18 hours. **(F)** Object-based 3D colocalization analysis (Imaris software) between α-syn-GFP aggregates and α-syn fibrils (74 ± 7%), α-syn-GFP aggregates and LAMP1+ lysosomes (69 ± 6%), and α-syn-GFP aggregates with both α-syn fibrils and LAMP1+ lysosomes (63 ± 6%). Mean ± SEM, $n = 2$ (16 SR images, approximately 2,000 lysosomes were analyzed). **(G)** SR image showing α-syn fibrils puncta and α-syn-GFP+ aggregate colocalizing with LAMP1 on the lysosomal membrane in CAD cells treated with α-syn fibrils for 1 hour. **(H)** SR image demonstrating α-syn fibrils puncta and α-syn-GFP+ aggregate colocalizing with LAMP1 both on the lysosomal membrane and inside the lysosomal lumen in CAD cells treated with α-syn fibrils for 2 hours. **(I)** SR image showing α-syn fibrils puncta and α-syn-GFP+ aggregate colocalizing with LAMP1 on the lysosomal membrane in CAD cells treated with α-syn fibrils for 4 hours. In C, D, E, G, H, and I insets of selected regions (white squares) from different α-syn-GFP transfected and α-syn fibril-treated CAD cells, labeled for DAPI and LAMP1 (pseudo colored in gray) are presented. Images were acquired with a spinning disk microscope with SR module. Scale bars: 10 μm (for insets: 1 μm; 0.5 μm for inset in G). The data underlying this figure may be found in S1 Data. α-syn, α-synuclein; CAD, Cath.a-differentiated; LAMP, lysosome-associated membrane protein; SR, super-resolution.

how propagation of misfolding occurs following the transfer of lysosomes containing α-syn aggregates to healthy neighboring cells [19,21]; and (ii) whether/how lysosome damage induced by α-syn aggregates favors the spread of pathology [19,32]. Lysosomes are dynamic organelles that move bidirectionally between the perinuclear and peripheral regions of the cell [90–92]. As our results indicate that the α-syn fibrils interfere with lysosome morphology and function, we hypothesized that they could also have an impact on lysosome positioning. Therefore, we evaluated the distribution of lysosomes in CAD cells challenged with α-syn fibrils compared to control cells. Cells were labeled for LAMP1, HCS CellMask Blue (cell outline stain), and DAPI (nuclear stain), and the number of lysosomes in the perinuclear and peripheral regions in each condition was quantified (see Materials and methods; Fig 5A). In agreement with previous findings that lysosomes are mainly located in the perinuclear region in healthy nonpolarized cells [93], we detected 57 ± 1% of the lysosomes at the perinuclear region in control CAD cells (Fig 5A, white arrows, and 5B). In contrast, in cells treated with α-syn fibrils, the situation was reversed, with the majority of lysosomes (56 ± 1%) found in the peripheral region (Fig 5A, green, red, yellow arrows, and 5B).

Since our results indicate that α-syn fibrils affect lysosomal pH and function (Fig 2B–2D), to understand whether there is a correlation between lysosome distribution and function, we next evaluated the distribution of lysosomes containing α-syn fibrils and quantified the perinuclear and peripheral distribution of only functional lysosomes positive for LysoTracker DR and Magic Red CathB. We found that α-syn fibril-containing LysoTracker DR positive and Magic Red CathB positive lysosomes were more perinuclear (54 ± 2% and 56 ± 1%, respectively; Fig 5C), suggesting that peripheral lysosomes containing α-syn fibrils are less functional.

These results thus show that α-syn fibrils promote redistribution of lysosomes toward the cell periphery. Furthermore, this pool of peripheral lysosomes is less acidic and has less degradative capacity.

## Peripheral α-syn–loaded lysosomes are preferentially transferred to neighboring cells

Our next question was to understand whether the effect of α-syn on lysosome positioning could be relevant for the propagation of the pathology and in which way. Peripheral lysosomes

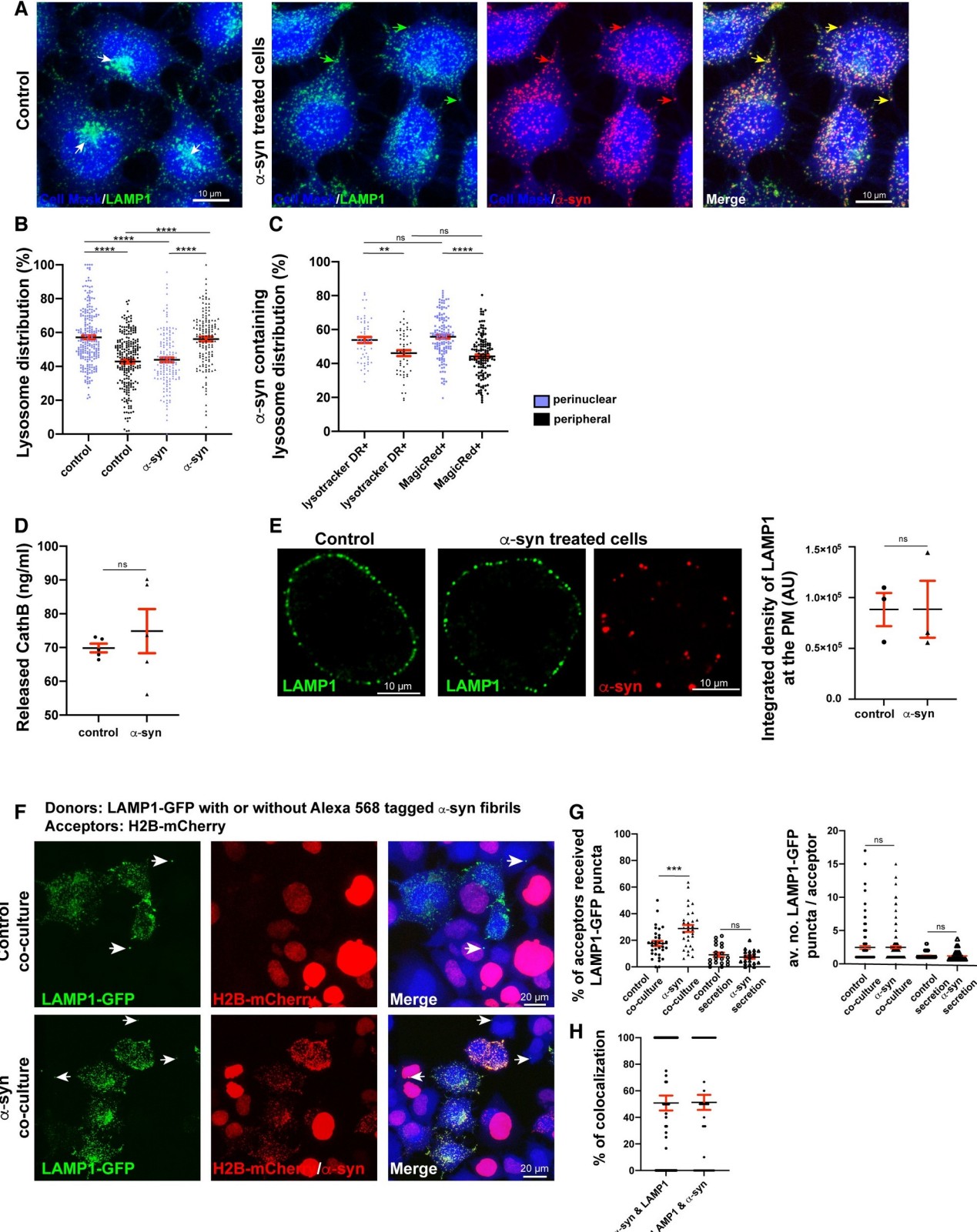

**Fig 5. α-Syn–loaded lysosomes localize to the cell periphery, are less functional, and are preferentially transferred to neighboring cells. (A)** Representative confocal images of control and Alexa 568–tagged α-syn fibril-treated CAD cells (18 hours) labeled for LAMP1-Alexa 488 antibody, HCS

CellMask Blue, and DAPI. White arrows indicate the perinuclear lysosome clusters in control cells; green, red, and yellow arrows indicate peripheral lysosomes, α-syn fibrils, and lysosomes colocalizing with α-syn fibrils, respectively. Scale bar: 10 μm. **(B)** % of LAMP1+ lysosomes in perinuclear (purple dots) and peripheral regions (black dots) of the control cells and cells challenged with α-syn fibrils is presented. In control cells: % of perinuclear lysosomes: 57 ± 1%, and peripheral lysosomes: 43 ± 1%; in α-syn fibril-treated cells: % of perinuclear lysosomes: 44 ± 1%, and peripheral lysosomes: 56 ± 1%. Mean ± SEM, $n$ = 5 (40 cells per condition). ns = not significant, $^{****}P < 0.0001$ by 1-way ANOVA followed by Tukey multiple comparison tests. **(C)** Graph representing the distribution of LysoTracker DR+ and Magic Red CathB+ lysosomes colocalizing with the α-syn fibrils in perinuclear (purple dots) and peripheral region (black dots). % of LysoTracker DR+ lysosomes colocalizing with α-syn fibrils in perinuclear region: 54 ± 2%, and in peripheral region: 46 ± 2%. % of Magic Red CathB+ lysosomes colocalizing with α-syn fibrils in perinuclear region: 56 ± 1%, and in peripheral region: 44 ± 1%. Mean ± SEM, $n$ = 4 (40 cells per condition). ns = not significant by 1-way ANOVA followed by Tukey multiple comparison tests. **(D)** Quantity of secreted CathB (ELISA) in the media of control (70 ± 1 ng/mL) and α-syn fibril-treated CAD cells (75 ± 7 ng/mL). Mean ± SEM, $n$ = 3. ns = not significant by Mann–Whitney $U$ test. **(E)** Representative confocal images of control and Alexa 568–tagged α-syn fibril-treated CAD cells for 18 hours, immuno-labeled for LAMP1-Alexa 488 antibody in non-permeabilized conditions. Scale bar: 10 μm. Average integrated density of LAMP1 immunofluorescence at the cell surface in control ($9 \times 10^4 \pm 2 \times 10^4$) and in α-syn fibril-treated cells ($9 \times 10^4 \pm 3 \times 10^4$) is presented. Mean ± SEM, $n$ = 3 (20 cells per condition). ns = not significant by paired $t$ test. Scale bar: 10 μm. **(F)** Representative confocal images of donor and acceptor cells after 24 hours of coculture. Donor cells were transfected with LAMP1-GFP and treated with Alexa 568–tagged α-syn fibrils or not (referred to as "α-syn coculture" and "control coculture," respectively). Acceptor cells were transfected with H2B-mCherry. Scale bar: 20 μm. **(G)** % of acceptor cells that received LAMP1+ puncta (lysosomes) from donor cells in control coculture (18 ± 2%), in α-syn coculture (29 ± 3%), in control secretion test (9 ± 2%), and in α-syn secretion test (7 ± 1%) (left graph). Mean ± SEM, $n$ = 4 (150 acceptor cells per condition). ns = not significant, $^{***}P = 0.0004$ by 1-way ANOVA followed by Tukey multiple comparison tests. The average number of LAMP1-GFP+ puncta detected per acceptor cell in control coculture (2 ± 0.2), in α-syn coculture (2 ± 0.2), in control secretion (1 ± 0.1), and in α-syn secretion (1 ± 0.1) is presented (right graph). Mean ± SEM, $n$ = 4 (150 acceptor cells per condition). ns = not significant, $^{***}P = 0.0004$ by Kruskal–Wallis nonparametric ANOVA test followed by Dunn multiple comparison tests. **(H)** % of α-syn fibril puncta colocalizing with LAMP1-GFP+ puncta (51 ± 6%) and % of LAMP1-GFP+ puncta colocalizing with α-syn fibrils puncta (51 ± 6%) is presented. Mean ± SEM, $n$ = 4 (20 acceptor cells that received both α-syn fibrils and LAMP1-GFP puncta were analyzed). The data underlying this figure may be found in S1 Data. α-syn, α-synuclein; AU, arbitrary unit; CAD, Cath.a-differentiated; CathB, Cathepsin B; LAMP, lysosome-associated membrane protein; LysoTracker DR, LysoTracker Deep Red.

could be more prone to exocytosis, and, therefore, could increase the secretion of α-syn fibrils to the extracellular environment. Alternatively, peripheral lysosomes could be more apt for transfer to healthy neighboring cells through TNTs. To distinguish between these alternative scenarios, we first examined lysosomal exocytosis by comparing the amount of CathB released into the culture medium of control and α-syn fibril-treated CAD cells using an ELISA method. These experiments showed no significant difference in the amount of CathB released by control (70 ± 1 ng/mL) and α-syn fibril-treated cells (75 ± 7 ng/mL; Fig 5D). As an alternative approach, we measured the integrated fluorescence density of LAMP1 at the PM in cells treated or not treated with α-syn fibrils. To this end, we used an antibody recognizing a luminal epitope of LAMP1, which is displayed to the extracellular milieu after fusion of lysosomes with the PM in non-permeabilized cells (Fig 5E). We did not observe a difference in the average integrated density of PM-associated LAMP1 between control ($9 \times 10^4 \pm 2 \times 10^4$ AU) and α-syn fibril-treated cells ($9 \times 10^4 \pm 3 \times 10^4$ AU; Fig 5E, graph). These results indicate that, despite their different distributions, lysosomes are exocytosed at a similar rate in α-syn fibril-treated and control cells.

Next, to understand whether the peripheral positioning of lysosomes could affect their transfer to neighboring cells, we compared the efficiency of lysosome transfer by using a coculture assay where donor cells were transiently transfected with LAMP1-GFP and either treated or not with α-syn fibrils for 18 hours (α-syn coculture and control coculture, respectively), and acceptor cells were transiently transfected with H2B-mCherry (Fig 5F, S3B Fig). We observed that the percentage of acceptor cells that received LAMP1-GFP puncta (lysosomes) from donor cells was significantly higher in the α-syn coculture compared to the control coculture (29 ± 3% and 18 ± 2%, respectively) (Fig 5G, left graph). Lysosome transfer through secretion was evaluated by a "secretion test" (S3I Fig), whereby acceptor cells treated with the conditioned medium of control or α-syn fibril-loaded donor cells were quantified for the presence of LAMP1-GFP puncta. We did not observe a significant difference between the 2 secretion tests (9 ± 2% and 7 ± 1% in control and α-syn secretion, respectively; Fig 5G, left graph). In addition, we did not observe any difference in the average number of LAMP1-GFP puncta

detected per acceptor cell either in the coculture conditions (2 ± 0.2 for both coculture conditions) or in the secretion test conditions (1 ± 0.1 for both secretion conditions; Fig 5G, right graph). These results confirm our previous data that lysosome transfer from donor to acceptor cells is predominantly mediated by cell-to-cell contact, likely through TNTs rather than secretion [19,21,22]. Importantly, we showed that donor cells challenged with α-syn fibrils transfer lysosomes more efficiently to the acceptor cells compared to control donor cells. Moreover, we performed object-based 2D colocalization analysis in the acceptor cells that received both LAMP1-GFP+ lysosomes and α-syn fibrils in order to evaluate the number of lysosomes containing α-syn fibrils transferred to the acceptor cells. We found 51 ± 6% of α-syn fibrils colocalizing with LAMP1-GFP+ lysosomes and 51 ± 6% of lysosomes colocalizing with α-syn fibrils, suggesting that half of the lysosomes transferred from donor to acceptor cells carried α-syn fibrils (Fig 5H).

Taken together, our data indicate that lysosomes that are more peripherally located in the presence of α-syn fibrils are not secreted more compared to control cells, but instead are more efficiently transferred to acceptor cells by cell-to-cell contact.

## TFEB as a player in lysosome redistribution upon α-syn loading

The TFEB regulates lysosome biogenesis and function by coordinating the expression of lysosomal components [94]. Furthermore, TFEB has also been shown to influence lysosome positioning [95–97]. To investigate a possible role of TFEB in the peripheral redistribution of lysosomes upon α-syn loading, we first asked whether the lysosomal loading of α-syn fibrils triggers TFEB nuclear translocation. To address this question, we expressed TFEB-wild type (WT)-GFP in CAD cells challenged or not with α-syn fibrils for 18 hours (S3A Fig, number 3). We found a significant increase in the percentage of CAD cells having nuclear TFEB upon α-syn loading compared to control cells (control: 18 ± 4%; α-syn: 57 ± 6%, Fig 6A).

Next, we assessed the effect of TFEB overexpression on lysosome distribution in CAD cells. We transiently transfected CAD cells with plasmids encoding WT TFEB (TFEB-WT-GFP) or constitutively active mutant TFEB (TFEB-mut-GFP) (i.e., nuclear TFEB) or with empty plasmid CMV-GFP (control) and quantified the number of lysosomes in the perinuclear and peripheral regions. We found that cells overexpressing the constitutively active form of TFEB showed more peripheral lysosomes compared to both control cells and cells overexpressing TFEB-WT-GFP (TFEB-mut-GFP: 59 ± 1%; TFEB-WT-GFP: 54 ± 2%; CMV-GFP: 52 ± 1%; Fig 6B). These results indicate that TFEB promotes the peripheral distribution of lysosomes in CAD cells.

To determine whether TFEB participates in peripheral redistribution of lysosomes upon α-syn loading, we silenced the expression of TFEB in CAD cells by RNA interference (Fig 6C). CAD cells transfected with scrambled small interfering RNA (siRNA; control) or siRNA targeting TFEB (siTFEB) were challenged with α-syn fibrils for 18 hours. Then, we quantified the positioning of lysosomes containing α-syn fibrils (Fig 6D). As expected, control cells showed more peripheral α-syn–loaded lysosomes (52.1 ± 1%). In contrast, in siTFEB-treated CAD cells α-syn–loaded lysosomes were slightly but significantly more perinuclear (50.7 ± 1%). These findings indicate that silencing of endogenous TFEB reduced the peripheral redistribution of lysosomes upon α-syn loading, suggesting the involvement of TFEB in this event.

## Lysosome positioning affects the efficiency of α-syn fibrils' transfer

To further investigate the effect of lysosome positioning on α-syn fibrils transfer, we set up different coculture assays where we monitored the amount of α-syn transfer using donor cells with more peripheral or more perinuclear lysosomes. To this end, we respectively

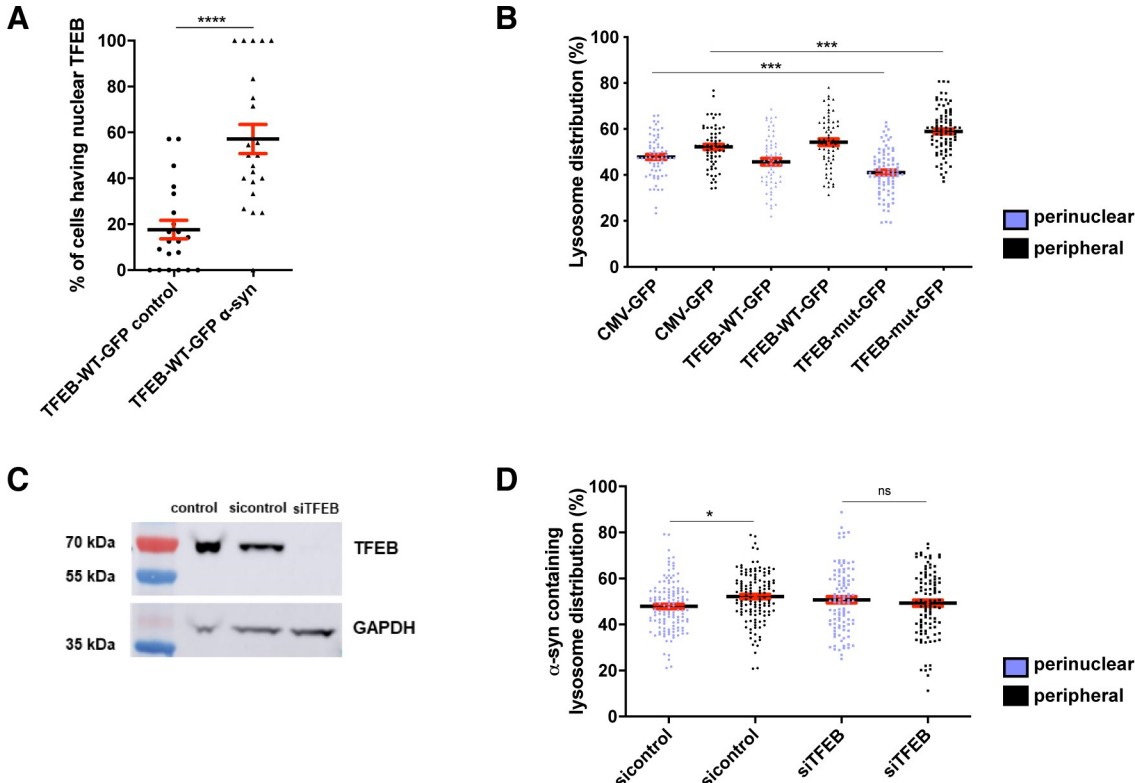

**Fig 6. The role of TFEB in lysosome redistribution upon α-syn loading. (A)** % of cells overexpressing TFEB-WT-GFP having nuclear TFEB in control CAD cells (18 ± 4%) and α-syn fibril-treated CAD cells for 18 hours (57 ± 6%). Mean ± SEM, $n$ = 3 (50 cells per condition). $^{****}P < 0.0001$ by unpaired $t$ test. **(B)** % of LAMP1+ lysosome distribution in perinuclear (purple dots) and peripheral regions (black dots) of cells overexpressing CMV-GFP (control), TFEB-WT-GFP, or TFEB-mut-GFP. In control CMV-GFP cells: % of perinuclear lysosomes: 48 ± 1%, and peripheral lysosomes: 52 ± 1%; in TFEB-WT-GFP cells: % of perinuclear lysosomes: 46 ± 2%, and peripheral lysosomes: 54 ± 2%; in TFEB-mut-GFP cells: % of perinuclear lysosomes: 41 ± 1%, and peripheral lysosomes: 59 ± 1%. Mean ± SEM, $n$ = 3 (30 cells per condition). $^{***}P < 0.001$ by 1-way ANOVA followed by Tukey multiple comparison tests. **(C)** Western blot prepared from cell extracts collected from control, scrambled siRNA (sicontrol)-, and siTFEB-pretreated CAD cells for 60 hours, showing TFEB and GAPDH (loading control) expression. **(D)** Graph representing the distribution of lysosomes containing α-syn fibrils in perinuclear (purple dots) and peripheral region (black dots) of α-syn fibril-treated sicontrol and siTFEB cells. % of lysosomes containing α-syn fibrils of sicontrol cells in perinuclear region: 49 ± 1%, and in peripheral region: 52 ± 1%. % of lysosomes containing α-syn fibrils of siTFEB cells in perinuclear region: 51 ± 1%, and in peripheral region: 49 ± 1%. Mean ± SEM, $n$ = 4 (30 cells per condition). ns = not significant, $^{*}P$ = 0.0288 by 1-way ANOVA followed by Tukey multiple comparison tests. The data underlying this figure may be found in S1 Data. α-syn, α-synuclein; CAD, Cath.a-differentiated; LAMP, lysosome-associated membrane protein; siRNA, small interfering RNA; siTFEB, siRNA targeting TFEB; sicontrol, scrambled siRNA; TFEB, transcription factor EB; WT, wild-type.

overexpressed or silenced the Arf-like small GTPase Arl8b, which controls the movement of lysosomes toward the cell periphery [98,99]. To examine the effect of peripheral lysosome redistribution, Arl8b-GFP transfected or control donor CAD cells were loaded with α-syn fibrils and cocultured with acceptor CAD cells transiently transfected with H2B-GFP to distinguish them from the donors (referred to as Arl8b-GFP coculture and control coculture, respectively; Fig 7A, S3C Fig). We then quantified the presence of α-syn fibrils in the acceptor cells in each coculture condition. We detected a significant increase in the percentage of acceptor cells that received α-syn fibrils in the Arl8b-GFP coculture (65 ± 4%) compared to control coculture (49 ± 4%; Fig 7B, left graph). As a control, we monitored the transfer of α-syn fibrils through the culture supernatant, thus checking a possible effect on transfer through a secretion mechanism. As previously found [19], the amount of α-syn transfer by secretion was very low

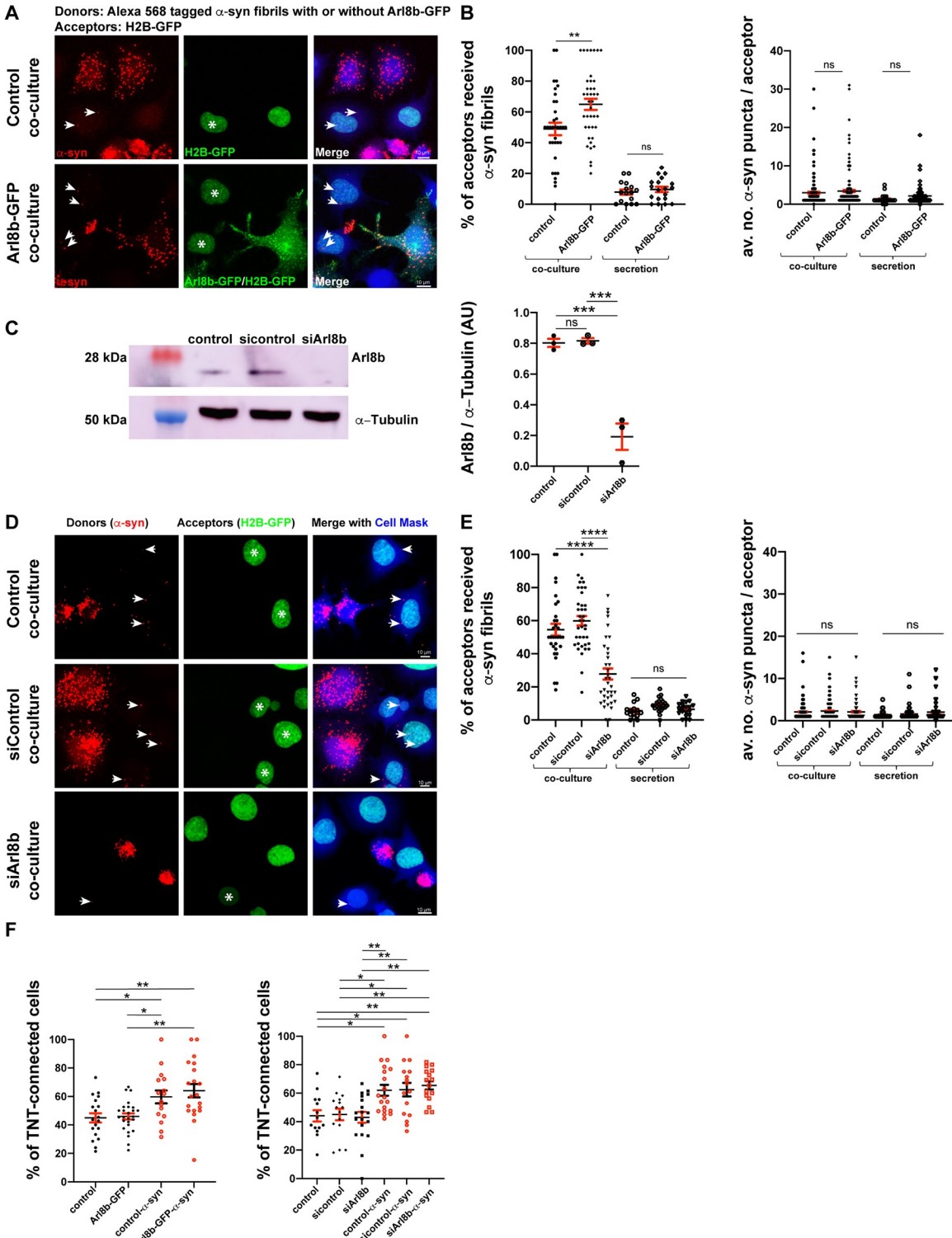

**Fig 7. Lysosome positioning affects the efficiency of α-syn fibrils' transfer. (A)** Representative confocal images of donor and acceptor cells after 24 hours of coculture. Donor cells were either loaded with Alexa 568–tagged α-syn fibrils only (control coculture) or transfected with Arl8b-GFP prior to be treated with α-syn fibrils (Arl8b-GFP coculture). Acceptor cells were transfected with H2B-GFP. Cocultures were labeled with HCS CellMask Blue and DAPI. Arrows indicate the α-syn fibrils received by acceptor cells, and asterisks indicate the cells having α-syn fibrils in each coculture condition. Scale bar: 10 μm. **(B)** % of acceptor cells received α-syn fibrils in control coculture (49 ± 4%), in Arl8b-GFP coculture (65 ± 4%), in control secretion (8 ± 2%), and in Arl8b-GFP secretion (10 ± 2%) (left

graph). Mean ± SEM, *n* = 3 (90 acceptor cells per condition). ns = not significant, **$P$ = 0.0038 by Kruskal–Wallis nonparametric ANOVA test followed by Dunn multiple comparison tests. Average number of α-syn fibrils puncta in control coculture (3 ± 0.4), in Arl8b-GFP coculture (3.4 ± 0.4), in control secretion (1.3 ± 0.4), and in Arl8b-GFP secretion (2 ± 0.3) (right graph). Mean ± SEM, *n* = 3 (90 acceptor cells per condition). ns = not significant, by 1-way ANOVA followed by Tukey multiple comparison tests. **(C)** Western blot prepared from cell extracts collected from control, scrambled siRNA (sicontrol)-, and siArl8b-pretreated CAD cells for 60 hours, showing Arl8b and α-tubulin (loading control) expression. Integrated density of Arl8b protein expression levels normalized to α-tubulin in control (0.80 ± 0.03), sicontrol- (0.82 ± 0.02), and siArl8b-pretreated cells (0.19 ± 0.09) are presented in AU. Mean ± SEM, *n* = 3. ns = not significant, ***$P$ = 0.0004 by 1-way ANOVA followed by Tukey multiple comparison tests. **(D)** Representative confocal images of donor and acceptor cells after 24 hours of coculture. Donor CAD cells were pretreated with sicontrol or siArl8b for 60 hours or left untreated prior to be challenged with Alexa 568–tagged α-syn fibrils (referred to as control, sicontrol, and siArl8b cocultures, respectively). Acceptor cells were transfected with H2B-GFP. Cocultures were labeled with HCS CellMask Blue and DAPI. Arrows indicate the α-syn fibrils received by acceptor cells, and asterisks indicate the cells having α-syn fibrils in each coculture condition. Scale bar: 10 μm. **(E)** % of acceptor cells that received α-syn fibrils in control coculture (55 ± 4%), sicontrol coculture (60 ± 3%), siArl8b coculture (28 ± 3%), control secretion (6 ± 1%), sicontrol secretion (9 ± 1%), and siArl8b secretion (6 ± 1%) (left graph). Mean ± SEM, *n* = 4 (80 acceptor cells per condition). ns = not significant, ****$P$ < 0.0001 by 1-way ANOVA followed by Tukey multiple comparison tests. Average number of α-syn puncta per acceptor cell in control coculture (2 ± 0.2), sicontrol coculture (2 ± 0.1), siArl8b coculture (2 ± 0.3), control secretion (1 ± 0.2), sicontrol secretion (2 ± 0.2), and siArl8b secretion (2 ± 0.3) (right graph). Mean ± SEM, *n* = 4 (80 acceptor cells per condition). ns = not significant by Kruskal–Wallis nonparametric ANOVA test followed by Dunn multiple comparison tests. **(F)** Control, Arl8b-GFP transfected, and sicontrol- or siArl8b-pretreated CAD cells for 60 hours were challenged or not with α-syn fibrils (18 hours) and labeled with WGA for TNT counting experiments. % of TNT-connected cells in control (45 ± 3%), Arl8b-GFP (46 ± 2%), control-α-syn (60 ± 5%), and Arl8b-GFP-α-syn (64 ± 5%) conditions. Mean ± SEM, *n* = 4 (115 cells per condition). ns = not significant (left graph). % of TNT-connected cells in control (44 ± 4%), sicontrol (45 ± 4%), siArl8b (43 ± 4%), control-α-syn (62 ± 4%), sicontrol-α-syn (62 ± 5%), and siArl8b-α-syn (66 ± 3%) is presented (right graph). Mean ± SEM, *n* = 3 (90 cells per condition). ns = not significant, *$P$ < 0.05, **$P$ < 0.01 by 1-way ANOVA followed by Tukey multiple comparison tests. The data underlying this figure may be found in S1 Data. α-syn, α-synuclein; AU, arbitrary unit; CAD, Cath.a-differentiated; siRNA, small interfering RNA; siArl8b, siRNA targeting Arl8b; sicontrol, scrambled siRNA; TNT, tunneling nanotube; WGA, wheat germ agglutinin.

compared with the transfer in coculture and did not significantly change in the 2 coculture conditions (8 ± 2% and 10 ± 2%, in control and Arl8b-GFP secretion conditions, respectively; Fig 7B, left graph). Despite changes in the percentage of cells that took up α-syn fibrils in cocultures, we did not find a significant difference in the average number of α-syn puncta per acceptor cell in any of these conditions (in average: 3 ± 0.4, 3.4 ± 0.4, 1.3 ± 0.4, and 2 ± 0.3 α-syn fibrils puncta per acceptor cell in control coculture, Arl8b-GFP coculture, control secretion, and Arl8b-GFP secretion, respectively; Fig 7B, right graph).

To corroborate these observations in a different cell type, we performed the same experiment in HeLa cells (S3D and S6 Figs) in which the effect of Arl8b on lysosome positioning was extensively characterized [90,91]. We observed that the percentage of acceptor cells that received α-syn fibrils was 28 ± 3% and 60 ± 5% in control and Arl8b-GFP cocultures, respectively (S6A and S6B Fig, left graph). Again, we did not observe any significant increase in the transfer of α-syn fibrils through secretion from the donors overexpressing Arl8b-GFP in comparison to control donors (8 ± 3% and 1 ± 1%, respectively; S6B Fig, left graph). As in CAD cells, the average number of α-syn puncta per acceptor cell was not significantly different in HeLa cocultures and secretion conditions (3 ± 1 and 5 ± 1 in control and Arl8b-GFP cocultures, respectively; 1 ± 0.0 and 2 ± 0.2 in control and Arl8b-GFP secretion tests, respectively; S6B Fig, right graph).

Furthermore, we silenced the expression of Arl8b in CAD cells by RNA interference (Fig 7C). Briefly, we pretreated donor cells with siRNA targeting Arl8b (siArl8b) and scrambled siRNA (sicontrol) for 60 hours prior to loading with α-syn fibrils for 18 hours; in parallel, untreated control cells were challenged with α-syn fibrils. These donors were then cocultured with acceptor cells transiently transfected with H2B-GFP (Fig 7D, S3E Fig). In control coculture and in sicontrol coculture, the percentage of acceptor cells that received α-syn fibrils was 55 ± 4% and 60 ± 3%, respectively, whereas in siArl8b coculture we observed a significant lower percentage of 28 ± 3% (Fig 7E, left graph). We did not detect any significant difference

in α-syn transfer through secretion (6 ± 1%, 9 ± 1%, and 6 ± 1% in control, sicontrol, and siArl8b secretion conditions, respectively; Fig 7E, left graph). The average number of α-syn puncta detected in acceptor cells in cocultures and secretion conditions was not significantly changed (1 to 2 puncta per cell in all the conditions; Fig 7E, right graph).

To further test the importance of lysosome positioning in α-syn fibrils transfer, we performed similar experiments in HeLa cells with knockout (KO) of components of the lysosome-positioning machinery. These components included the myrlysin (also known as BORCS5) subunit of BORC, which promotes recruitment of Arl8b to membranes [100], and the kinesins KIF1B and KIF5B [101–105], which drive lysosome movement toward the cell periphery [90]. KO of these proteins impairs lysosome movement toward the cell periphery [90,100]. These KO donor cells, in parallel to control donors (WT HeLa), were loaded with α-syn fibrils and cocultured with WT HeLa cells expressing H2B-GFP (S3F and S6C Figs). Acceptor cells were then analyzed for the presence of α-syn fibrils. The percentage of acceptor cells that received α-syn fibrils in control coculture was 29 ± 2%, whereas in myrlysin-KO and KIF1B-KIF5B-double-KO (KIF1B-5B dKO) cocultures, this percentage was significantly decreased to 16 ± 2% and 12 ± 2%, respectively (S6D Fig, left graph). α-Syn fibrils' transfer through secretion was similar in all the conditions (5 ± 2%, 6 ± 1%, and 6 ± 1% in control, myrlysin-KO, and KIF1B-5B dKO secretion conditions, respectively; S6D Fig, left graph). We did not observe a significant difference in the average number of transferred α-syn puncta in any of the cocultures/secretion conditions (1 to 2 puncta per cell in all the conditions; S6D Fig, right graph).

Overall, these data indicate that the contact-dependent transfer of α-syn is affected by the position of lysosomes, with more transfer when lysosomes are peripheral and less transfer when lysosomes are perinuclear.

As we have shown that lysosomes containing α-syn are transferred from acceptor cells inside TNTs [52] (S2 Fig), we wondered whether TNT formation was affected by the manipulations that alter lysosome positioning. Thus, we performed 2 sets of TNT counting experiments in control, Arl8b-GFP–overexpressing, or siArl8b/sicontrol-treated CAD cells in the presence or absence of α-syn fibrils. These cells were labeled with wheat germ agglutinin (WGA), and the percentage of TNT-connected cells was quantified for each condition (see Materials and methods). We did not observe a significant difference in the percentage of TNT-connected cells between control (45 ± 3%) and Arl8b-GFP–overexpressing cells (46 ± 2%; Fig 7F, left graph). However, in the presence of α-syn fibrils, the percentage of TNT-connected cells was significantly increased to 60 ± 5% in control and 64 ± 5% in Arl8b-GFP–overexpressing cells (Fig 7F, left graph). Similarly, the percentage of TNT-connected cells remained unchanged between control (44 ± 4%), sicontrol (45 ± 4%), and siArl8b (43 ± 4%) pretreated cells, whereas in the presence of α-syn fibrils, the percentage of TNT-connected cells significantly increased to 62 ± 4% in control, 62 ± 5% in sicontrol, and 66 ± 3% in siArl8b-treated cells (Fig 7F, right graph). These data indicate that lysosome positioning has no effect on TNT formation per se, but it is the presence of α-syn fibrils that increases the number of TNT-connected cells, in line with previous findings [19].

Taken together, these results indicate that lysosome positioning affects the efficiency of α-syn fibrils' transfer. Moreover, α-syn fibrils increase the number of TNT-connected cells independent of lysosome positioning.

## α-Syn fibrils induce LMP upon arrival in the acceptor cells

Our data show that more than 50% of α-syn fibrils in acceptor cells colocalize with LAMP1, indicating that they are still associated with lysosomes after the transfer (Fig 5H). Moreover,

we have previously reported that α-syn fibrils can induce seeding of soluble α-syn once they are transferred to acceptor cells [19]. Based on the results presented here, one possible explanation for this observation would be that α-syn fibrils also trigger LMP in acceptor cells following their transfer, similar to what we have observed in donor cells (Fig 3A and 3B). To assess this hypothesis, we performed coculture experiments where donor cells were either transiently transfected with H2B-mCherry (control coculture) or treated with α-syn fibrils for 18 hours (α-syn coculture) and cocultured with acceptor cells transiently transfected with Gal3-GFP (Fig 8A, S3G Fig, number 1). Acceptor cells were then evaluated for the presence of Gal3-GFP puncta. The percentage of acceptor cells exhibiting LMP (Fig 8A, white arrows) was significantly higher in the α-syn coculture (33 ± 5%) compared to control coculture (6 ± 2%) (Fig 8B, upper graph). We also quantified the average number of Gal3-GFP puncta per acceptor cell, and we found a significant increase in α-syn coculture (14 ± 1) compared to control coculture where some LMP occurs spontaneously (8 ± 1; Fig 8B, lower graph). We also performed SR microscopy in α-syn coculture labeled for lysosomes with antibody to LAMP1 and observed Gal3-GFP and α-syn fibrils puncta both in close proximity inside the lysosomal lumen (Fig 8C) and colocalizing on the lysosomal membrane (Fig 8D). These results thus show that, following their transfer, α-syn–containing lysosomes undergo LMP in acceptor cells.

### α-Syn fibrils induce seeding after their transfer into acceptor cells, and this event is preferentially associated with lysosomes

To further investigate the seeding mechanism in acceptor cells that received lysosomes containing α-syn fibrils, we used again our α-syn-GFP soluble fluorescent reporter expressed in acceptor cells cocultured with donor cells, either challenged with α-syn fibrils or transfected with H2B-mCherry (α-syn coculture and control coculture, respectively; Fig 9A, S3G Fig, number 2). We then quantified the presence of α-syn-GFP puncta (indicating aggregation of the soluble protein; Fig 9A, white arrows) in the acceptor cells that received α-syn fibrils (Fig 9A, red arrows). We found that 27 ± 4% of acceptor cells contained 7 ± 1 α-syn-GFP puncta in α-syn coculture, whereas in control coculture, only 10 ± 4% of the acceptor cells had 2 ± 0.2 α-syn-GFP puncta (Fig 9B).

Next, to address whether the seeding in acceptor cells was associated with lysosomes, we performed SR microscopy. After labeling the cells in coculture with antibody to LAMP1, we examined the subcellular localization of α-syn-GFP aggregates in acceptor cells. We observed 4 different scenarios: (i) lysosomes containing α-syn fibrils and colocalizing with α-syn-GFP puncta surrounding the lysosomal membrane (Fig 9C); (ii) lysosomes filled with α-syn fibrils with discrete α-syn-GFP deposits on their membrane (Fig 9D); (iii) lysosomes containing α-syn-GFP puncta with α-syn fibrils in close proximity (Fig 9E); and (iv) lysosomes containing both α-syn fibrils and α-syn-GFP puncta partially colocalizing in their lumen (Fig 9F). In addition, we performed 3D object-based colocalization analysis between α-syn-GFP puncta, α-syn fibrils, and LAMP1+ lysosomes. We observed that 60 ± 7% of the newly formed α-syn-GFP aggregates colocalized with LAMP1+ lysosomes (Fig 9G). Overall, these data indicate that the majority of the seeding of endogenously expressed soluble α-syn takes place at lysosomal compartments in acceptor cells after the transfer of α-syn fibrils.

### α-Syn–loaded cells transfer damaged lysosomes to healthy cells and receive healthy lysosomes in return

In our homotypic coculture system between CAD cells, we could not address the nature of the transferred lysosomes (i.e., whether the lysosomes that are transferred have LMP or not) nor

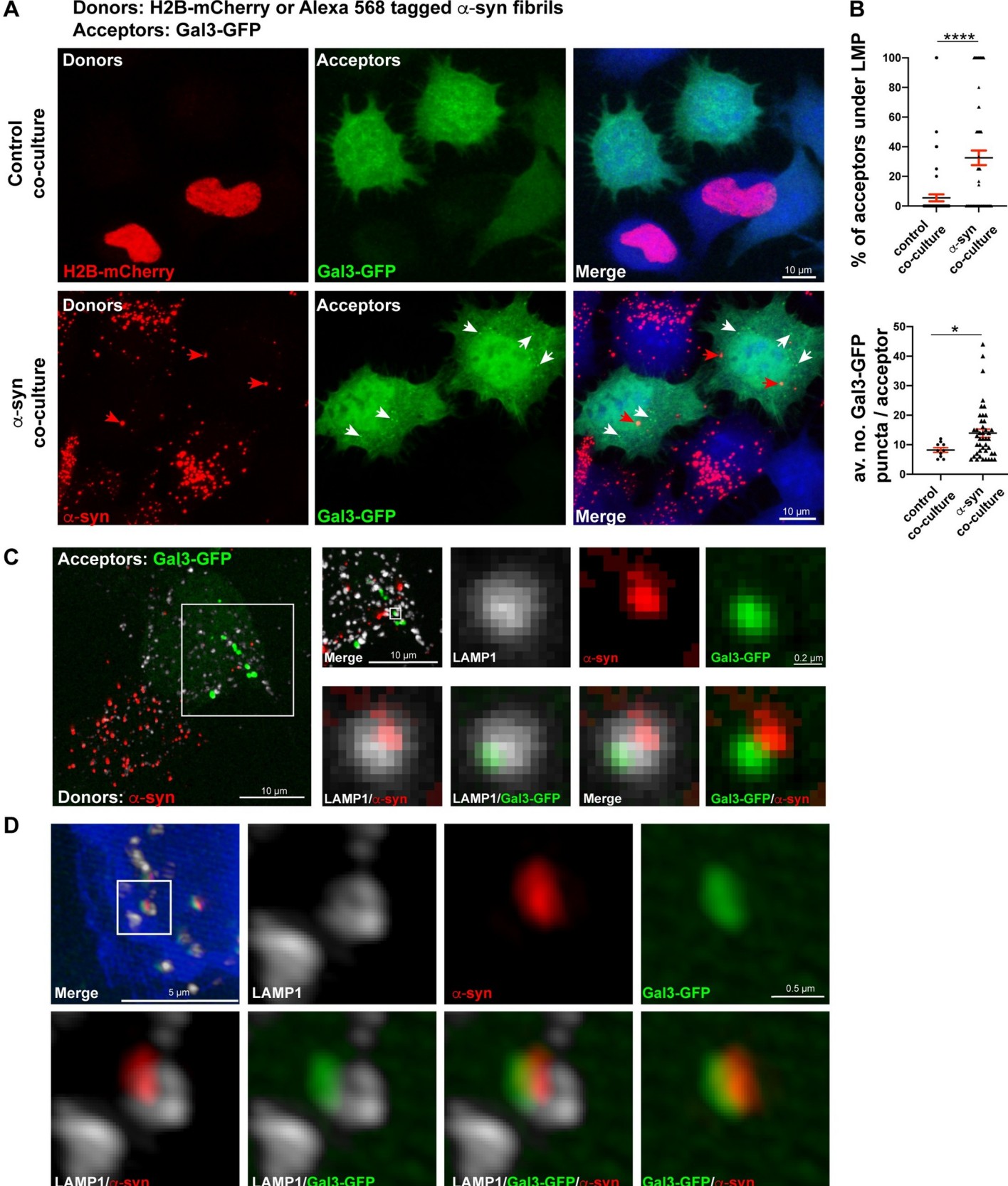

**Fig 8. α-Syn fibrils induce LMP upon arrival in the acceptor cells. (A)** Representative confocal images of donor and acceptor cells after 24 hours of coculture. Donor cells were transfected with H2B-mCherry (control coculture) or treated with Alexa 568–tagged α-syn fibrils (α-syn coculture). Acceptor cells were transfected with Gal3-GFP. Cocultures were labeled with HCS CellMask Blue. Red arrows indicate the α-syn fibrils received by acceptor cells, white arrows indicate the Gal3-GFP puncta formation (LMP) in the acceptor cells that received α-syn fibrils. Scale bar: 10 μm. **(B)** % of acceptor cells under LMP in control (6 ± 2%) and α-syn (33 ± 5%) cocultures (upper graph). Average number of Gal3-GFP puncta formation in acceptor cells under LMP in control (8 ± 1) and α-syn (14 ± 1) cocultures (lower graph). Mean ± SEM, $n = 5$ (50 acceptor cells per condition). $^*P = 0.04$, $^{****}P < 0.0001$ by Mann–Whitney $U$ test. **(C)** SR image of a coculture where donor cells were treated with α-syn fibrils for 18 hours and acceptor cells were transfected with Gal3-GFP, labeled for LAMP1-Alexa 647 antibody (pseudo colored in gray). A selected region where Gal3-GFP puncta were observed (indicated by a white square in the acceptor cell of the coculture image) was further magnified to observe one lysosome (indicated by the small white square) having α-syn fibrils puncta and recruited Gal3-GFP. Image was acquired with a spinning disk microscope with SR module. Scale bar: 10 μm (for the inset: 0.2 μm). **(D)** SR image of an acceptor cell demonstrating Gal3-GFP puncta recruited at the lysosomal membrane where α-syn fibrils were detected in close proximity. Selected lysosome is indicated by a white square. Image was acquired by SR SIM (Zeiss, LSM 780-Elyra PS.1). Scale bar: 5 μm (for the inset: 0.5 μm). In C, the signal was boosted to make the selected lysosome visible; in D, the acceptor cell was presented with HCS CellMask Blue labeling as this cell was poorly transfected but accumulation of Gal3-GFP puncta was clearly detectable. The data underlying this figure may be found in S1 Data. α-syn, α-synuclein; Gal3, Galectin-3; LAMP, lysosome-associated membrane protein; LMP, lysosomal membrane permeabilization; SIM, structured illumination microscopy; SR, super-resolution.

the directionality of lysosome transfer (from donor to acceptor and/or from acceptor to donor). To answer these fundamental questions, we designed a heterotypic coculture assay using mouse CAD cells as donors and human HeLa cells as acceptors. Briefly, CAD cells were transiently transfected with Gal3-GFP, loaded or not with α-syn fibrils, and used as the donor cell population; on the other hand, HeLa cells stably expressing Gal3-Turquoise were used as the acceptor cell population. Donor and acceptor cells were cocultured for 24 hours prior to be fixed and immunolabeled with either a specific anti-mouse LAMP1 antibody (to detect the lysosomes originated from donor cells) or a specific anti-human LAMP1 antibody (to detect the lysosomes originated from acceptor cells; S3H Fig).

First, we tested this heterotypic system by evaluating (i) the efficiency of α-syn fibrils' transfer from donor CAD cells to acceptor HeLa cells; and (ii) the occurrence of seeding in the acceptor cells. The percentage of acceptor HeLa cells that received α-syn fibrils from donors was 65 ± 3% with an average of 4 ± 0.2 α-syn puncta per acceptor cell (S7A and S7B Fig). As expected, also in this case, both the percentage of α-syn fibrils (7 ± 2%) and the average number of α-syn fibrils puncta per acceptor cell (2 ± 0.2) was significantly lower in the secretion condition (S7B Fig). When donor CAD cells loaded with α-syn fibrils were cocultured with HeLa cells transiently transfected with soluble α-syn-GFP, we again observed the aggregation of soluble α-syn-GFP (S7C Fig, green arrows) in acceptor HeLa cells that received α-syn fibrils from donor CAD cells (S7C Fig, red arrows). Taken together, these results indicate that α-syn fibrils can be efficiently transferred in a cell-to-cell contact manner and can induce seeding in the acceptor cells following their transfer also in a heterotypic mouse–human coculture assay.

As we could recognize the origin of lysosomes, we then used this system to better quantify the transfer of all lysosomes and of lysosomes under LMP from donor to acceptor cells. To this end, CAD cells were either loaded or not with α-syn fibrils and cocultured with HeLa cells expressing Gal3-Turquoise. Cells in coculture were then immunolabeled with the antibody to mouse LAMP1 to detect the transfer of lysosomes originated from donor cells (Fig 10A). The percentage of acceptor HeLa cells that received donors' lysosomes was significantly higher in α-syn coculture (52 ± 4%) compared to control coculture (32 ± 5%), whereas the average number of transferred lysosomes (LAMP1+ puncta) did not change (3 ± 0.3 in both cocultures; Fig 10B). Furthermore, we observed that in α-syn coculture, both lysosomes containing α-syn fibrils (Fig 10A, yellow arrows; S7G Fig, red arrow) or not (Fig 10A, white arrows; S7G Fig, white arrow) were transferred to acceptor cells.

Next, to evaluate the amount of lysosomes having LMP that were transferred from CAD cells to HeLa cells, we quantified the occurrence of LMP in CAD cells in our heterotypic coculture condition. Thus, we counted the number of Gal3-GFP puncta in donor CAD cells transfected with Gal3-GFP, challenged or not with α-syn fibrils, and cocultured with

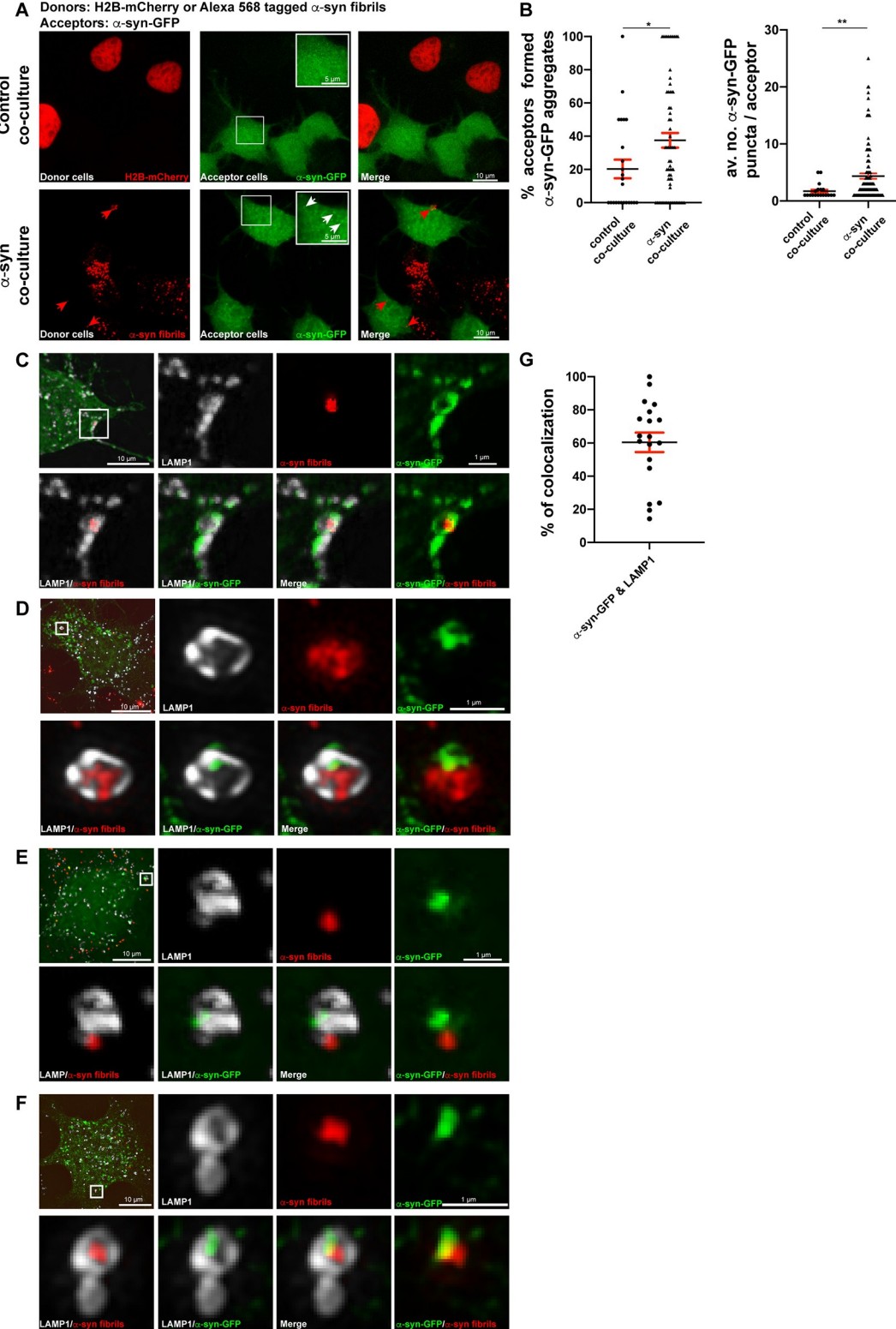

**Fig 9. α-syn fibrils induce seeding after their transfer into acceptor cells, and this event is frequently associated with lysosomes.** **(A)** Representative confocal images of donor and acceptor cells after 24 hours of coculture. Donor cells were transfected with H2B-mCherry (control coculture) or treated with Alexa 568–tagged α-syn fibrils for 18 hours (α-syn coculture). Acceptor cells were transfected with α-syn-GFP. Inset of acceptor cells (indicated by white squares) is presented at the right top corner of acceptor cells' panel. White arrows indicate α-syn-GFP aggregate formation, and red arrows

indicate α-syn fibrils received by acceptor cells in α-syn coculture. Scale bar: 10 μm (for the inset: 5 μm). **(B)** % of acceptor cells that formed α-syn-GFP aggregates in control (10 ± 4%) and in α-syn (27 ± 4%) cocultures (left graph). Average number of α-syn-GFP puncta detected in acceptor cells in control (2 ± 0.2) and in α-syn (7 ± 1) cocultures is presented (right graph). Mean ± SEM, $n$ = 3 (150 acceptor cells per condition). *$P$ = 0.01 and **$P$ = 0.003 for % of aggregates and average number of aggregates in control versus α-syn cocultures, respectively, by Mann–Whitney $U$ test. **(C)** SR image showing a lysosome containing α-syn fibrils puncta colocalizing with α-syn-GFP aggregate at the lysosomal membrane. **(D)** SR image showing a lysosome containing α-syn fibrils where α-syn-GFP+ aggregate is located mostly in a restricted portion of the lysosomal membrane. **(E)** SR image showing a lysosome having α-syn-GFP+ aggregate where α-syn fibrils puncta were found to be in close proximity of the lysosomes. **(F)** SR image showing a lysosome where both α-syn fibrils and α-syn-GFP aggregates are located to the lumen of the lysosome and partially colocalizing. **(G)** Object-based 3D analysis (Imaris software) of colocalization between α-syn-GFP and LAMP1+ lysosomes (60 ± 7%). Mean ± SEM, $n$ = 2 (15 SR images, approximately 3,000 lysosomes were analyzed). Insets of images in C and D (acquired with spinning disk microscope with SR module) and E and F (acquired with SR SIM, Zeiss, LSM 780-Elyra PS.1) from 4 selected regions (white squares) and from different acceptor cells transfected with α-syn-GFP and that received α-syn fibrils, labeled for LAMP1 Alexa 647 antibody (pseudo colored in gray). Scale bars: 10 μm (for the insets: 1 μm). The data underlying this figure may be found in S1 Data. α-syn, α-synuclein; LAMP, lysosome-associated membrane protein; SIM, structured illumination microscopy; SR, super-resolution.

HeLa-Gal3-Turquoise cells (S7D Fig, green arrows). As expected, the percentage of donor CAD cells having LMP was significantly higher in α-syn loaded donor cells (11 ± 4%) compared to control donors (2 ± 2%), whereas the average number of Gal3-GFP puncta per cell did not show any statistically significant change (10 ± 2 and 14 ± 4 in control and α-syn donors, respectively; S7E Fig). These findings indicate that, in coculture conditions, donor CAD cells challenged with α-syn fibrils undergo LMP similar to what we showed earlier (Fig 3A and 3B). Next, we evaluated the transfer of donors' lysosomes with LMP by quantifying the presence of Gal3-GFP puncta in acceptor HeLa-Gal3-Turquoise cells both in control and α-syn cocultures (Fig 10C). We found a significantly higher percentage of HeLa cells with Gal3-GFP puncta in the α-syn coculture (13 ± 2%; Fig 10C, green arrows, and D, left graph) compared to control coculture (4 ± 1%; Fig 10D, left graph). We did not observe a significant difference between the average number of Gal3-GFP puncta per acceptor cell in the 2 cocultures (2 ± 0.2 and 2 ± 0.3 in control and α-syn cocultures, respectively; Fig 10D, right graph). These data indicate that CAD cells challenged with α-syn fibrils transfer more damaged lysosomes (i.e., undergoing LMP) to acceptor cells compared to control CAD cells. In addition, we were able to detect donors' lysosomes colocalizing both with α-syn fibrils and Gal3-GFP puncta in acceptor HeLa cells (Fig 10E), demonstrating that damaged lysosomes carrying α-syn fibrils are transferred into the acceptor cells.

By taking advantage of HeLa cells stably expressing Gal3-Turquoise as acceptor cells, we also evaluated the occurrence of LMP in acceptor cells by assessing the Gal3-Turquoise puncta (Fig 10F, turquoise arrows). We found that 18 ± 3% of acceptor cells undergo LMP with an average of 10 ± 1 Gal3-Turquoise puncta per cell in α-syn coculture (Fig 10G). LMP in the acceptor cells of control coculture was significantly lower (3 ± 1%) with an average of 9 ± 1 Gal3-Turquoise puncta per cell (Fig 10G). In addition, by using antibody to human LAMP1, we also detected colocalization with Gal3-Turquoise puncta in acceptor cells (S7F Fig, arrow). These results suggest that α-syn fibrils derived from CAD cells are able to induce LMP in the acceptor HeLa cells following their transfer. Interestingly, in very few cases, we could also detect colocalization between Gal3-Turquoise puncta and donors' lysosomes labeled with anti-mouse LAMP1 antibody in acceptor HeLa cells (S7G Fig, white arrow), suggesting that damaged donors' lysosomes can undergo LMP upon arrival to the acceptor cells.

Finally, to evaluate lysosome transfer from acceptor to donor cells, the CAD-HeLa coculture was immunolabeled with anti-human LAMP1 antibody, and donor CAD cells were evaluated for the presence of acceptors' lysosomes (Fig 10H). The percentage of donor cells that received acceptors' lysosomes in control coculture was 14 ± 4%, whereas in α-syn coculture, this percentage was significantly higher (34 ± 5%; Fig 10I, left graph). The average number of

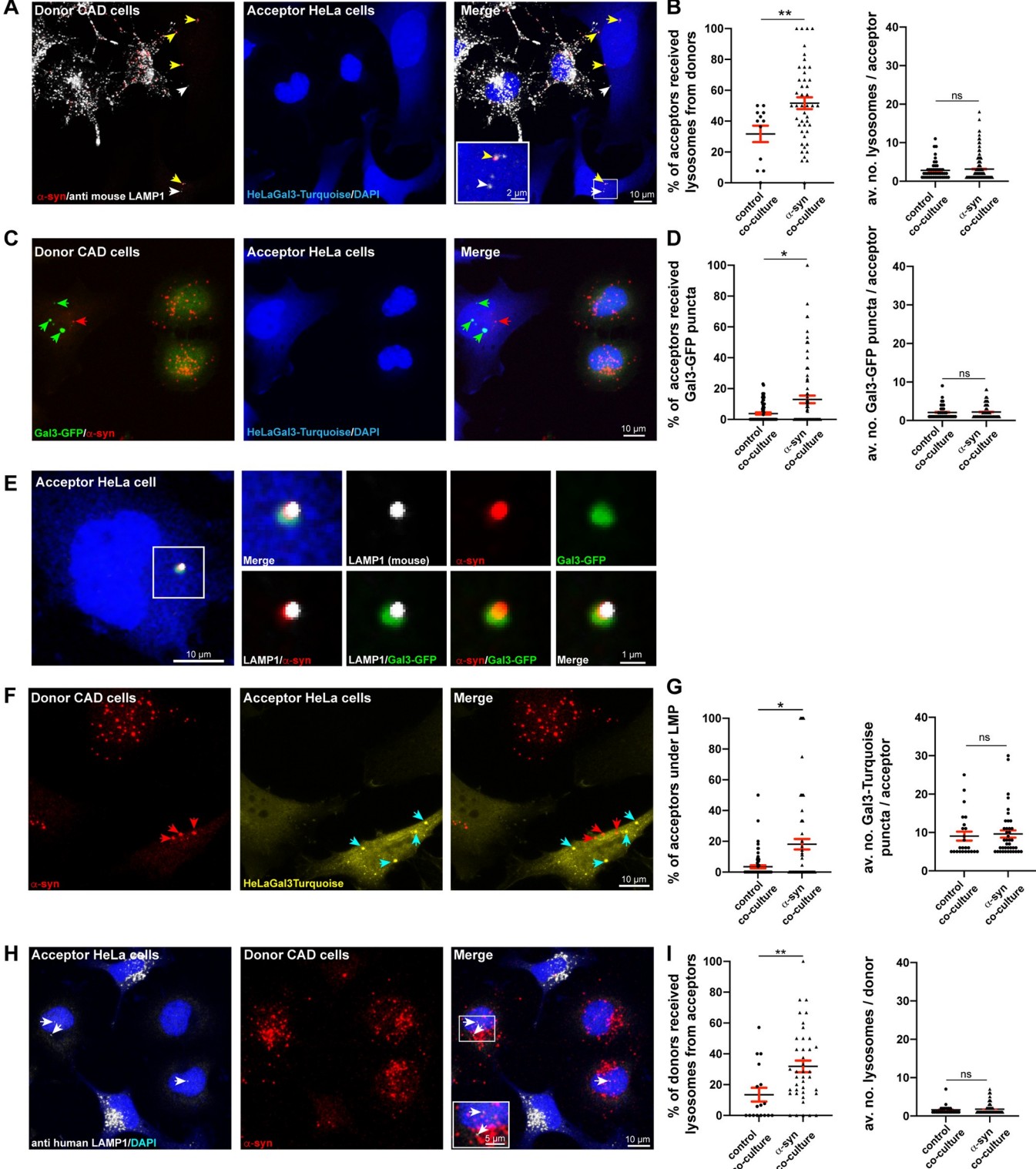

**Fig 10. Intercellular lysosome transfer is reciprocal. (A)** Representative confocal image of CAD-HeLa coculture where donor mouse CAD cells were treated with Alexa 568–tagged α-syn fibrils (18 hours) and cocultured with acceptor human HeLa stable Gal3-Turquoise cells for 24 hours, labeled for anti-LAMP1 mouse antibody and DAPI. In acceptor HeLa Gal3-Turquoise cells, arrows indicate donors' lysosomes containing (yellow) or not (white) α-syn fibrils. Inset represents a selected region (white square) of one acceptor cell containing donors' lysosomes. Scale bar: 10 μm (for the inset: 2 μm). **(B)** % of acceptor HeLa cells that received lysosomes from donor CAD cells in control (32 ± 5%) and in α-syn (52 ± 4%) cocultures (left graph). **P = 0.006 by 2-tailed *t* test. Average number of lysosomes transferred from donor to acceptor cells in control (3 ± 0.3) and in α-syn (3 ± 0.3) cocultures (right graph). Mean ± SEM. ns = not

significant by Mann–Whitney *U* test. **(C)** Representative confocal image of CAD-HeLa coculture where Gal3-GFP transfected donor CAD cells were treated with Alexa 568–tagged α-syn fibrils for 18 hours and cocultured with HeLa Gal3-Turquoise for 24 hours, labeled with DAPI. In acceptor HeLa Gal3-Turquoise cells, red arrows indicate α-syn fibrils, and green arrows indicate Gal3-GFP puncta (lysosomes under LMP received from donor cells). Scale bar: 10 μm. **(D)** % of acceptor HeLa cells that received Gal3-GFP puncta from donor cells in control (4 ± 1%) and in α-syn (13 ± 2%) cocultures (left graph); average number of Gal3-GFP puncta per acceptor HeLa cell in control (2 ± 0.3) and in α-syn (2 ± 0.2) cocultures (right graph). Mean ± SEM. ns = not significant, *P* = 0.04 by Mann–Whitney *U* test. **(E)** Acceptor HeLa Gal3-Turquoise cell that received a donor's lysosome containing α-syn fibrils and under LMP. Colocalization between LAMP1 (labeled with anti-LAMP1 mouse antibody), α-syn fibrils, and Gal3-GFP is presented in the inset of the selected area (white square). Scale bar: 10 μm (for the inset: 1 μm). **(F)** Representative confocal image of LMP in HeLa Gal3-Turquoise (pseudo colored in yellow) acceptor cell in CAD-HeLa coculture. In acceptor HeLa Gal3-Turquoise cells, red arrows indicate the α-syn fibrils, and turquoise arrows indicate the Gal3Turquoise puncta. Scale bar: 10 μm. **(G)** % of acceptor HeLa cells under LMP in control (3 ± 1%) and in α-syn (18 ± 3%) cocultures is presented (left graph); average number of Gal3-Turquoise puncta per acceptor cell under LMP in control (9 ± 1) and in α-syn (10 ± 1) cocultures is presented (right graph). Mean ± SEM. ns = not significant, *P* = 0.03 by Mann–Whitney *U* test. **(H)** Representative confocal image of CAD-HeLa coculture where donor CAD cells were treated with Alexa 568–tagged α-syn fibrils for 18 hours and cocultured with HeLa Gal3-Turquoise for 24 hours, labeled for anti-LAMP1 human antibody and DAPI. In donor CAD cells, white arrows indicate acceptors' lysosomes. Inset of a selected region of a donor CAD cell (white square) showing acceptors' lysosomes in higher magnification. Scale bar: 10 μm (for the inset: 5 μm). **(I)** % of acceptors' lysosomes transferred from acceptor to donor cells in control (14 ± 4%) and in α-syn (34 ± 5%) cocultures (left graph). **P = 0.002 by 2-tailed *t* test. Average number of acceptors' lysosomes per donor cell in control (2 ± 0.3) and in α-syn (2 ± 0.1) cocultures (right graph). Mean ± SEM. ns = not significant by Mann–Whitney *U* test. In CAD-HeLa coculture model, data were collected from 7 independent experiments in which 4 and 3 of them were labeled with anti-LAMP1 mouse and human antibodies, respectively. According to the experiment, in average, 100 acceptor or donor cells were analyzed for each condition per experiment. The data underlying this figure may be found in S1 Data. α-syn, α-synuclein; CAD, Cath.a-differentiated; Gal3, Galectin-3; LAMP, lysosome-associated membrane protein; LMP, lysosomal membrane permeabilization.

acceptors' lysosomes per donor cell in both coculture conditions was similar (2 ± 0.3 and 2 ± 0.1 in control and α-syn cocultures, respectively; Fig 10I, right graph). To understand whether lysosomes transferred from acceptor to donor cells were damaged, we quantified the presence of Gal3-Turquoise puncta in donor CAD cells in both in the α-syn and control cocultures (S7H Fig). Only 4 ± 1% of donor cells resulted positive for Gal3-Turquoise puncta in α-syn coculture, while we could barely (0.3 ± 0.3%) observe donor cells positive for Gal3-Turquoise puncta in control coculture (S7I Fig). Taken together, these results demonstrate that donor cells challenged with α-syn fibrils receive healthy lysosomes from acceptor cells, while the transfer of damaged lysosomes from the acceptor cells is a rare event. This suggests the occurrence of a rescue mechanism toward donor cells overloaded with α-syn, similar to what was previously reported in the case of a lysosomal disease [106].

## Discussion

α-Syn plays a crucial role in the generation and progression of synucleinopathies including PD. Indeed, the pathology correlates with the presence of α-syn inclusions spreading through interconnected brain regions in a "prion-like" manner [5,16,107]. By using a coculture system, we previously demonstrated that α-syn fibrils can be transferred from donor to acceptor cells inside of lysosomes through TNTs and induce the seeding in the recipient cells [19]. However, fundamental questions regarding how lysosomes participate in this process remained open. In the current study, we focused on the role of lysosomes and the crosstalk between α-syn fibrils and lysosomes in the context of intercellular spread of α-syn fibrils and the subsequent seeding mechanism.

By using SR microscopy, we looked precisely at the subcellular localization of the α-syn fibrils, and we found them mainly in the lysosomal lumen or at the lysosomal membrane (Fig 1B). In both cases, lysosomes associated with α-syn fibrils had a larger diameter compared to control lysosomes. Lysosomal enlargement in the presence of α-syn fibrils has been reported before [70,108]; however, here, we showed that this morphological change is specific to the lysosomes containing α-syn fibrils. Indeed, in α-syn fibril-treated cells, the diameter of the lysosomes devoid of fibrils was comparable to that of the control lysosomes (Fig 1B, graph). This finding was supported by correlative EM analysis that indicated an α-syn-dependent enlargement of lysosomes both in the perimeter and area (Fig 1D, graphs). In addition, we

found that lysosomes containing α-syn fibrils exhibit structural alterations (e.g., enlargement in size and aberrant luminal content), directly supporting that α-syn fibrils cause changes in lysosome morphology (Fig 1C). The exact mechanism of lysosomal enlargement needs to be further investigated; in agreement with studies demonstrating lysosome dilation accompanied by lysosomal dysfunction [70,108,109], one possibility is that dysfunction results from accumulation of undegraded α-syn fibrils. Consistently, while the total number of lysosomes did not vary, we detected lysosomal dysfunction in α-syn fibril-treated cells compared to control cells. Specifically, in the presence of α-syn fibrils, we observed disturbance of lysosomal pH and decrease of lysosomal enzyme activities (Fig 2B and 2D). These changes could be a consequence of the LMP induced by α-syn fibrils that we and others detected in α-syn fibril-treated cells [73,74]. Indeed, LMP causes the release of cathepsins and other hydrolases from the lysosomal lumen to the cytosol [72,110].

LMP can be induced by different protein aggregates (e.g., α-syn, tau, and huntingtin), and it has been proposed to be a common mechanism used by exogenous fibrils to spread through the cells [74]. Specifically, LMP has been described in several cell types challenged with α-syn fibrils, such as the neuronal cell lines SH-SY5Y and N27, as well as human dopaminergic neurons derived from induced pluripotent stem cells [73,74]. Here, we showed that α-syn fibrils induce LMP also in neuronal CAD cells (Fig 3A and 3B). By using SIM, we observed the recruitment of Gal3 to the damaged lysosomes displaying α-syn fibrils in their lumen or limiting membrane (Fig 3D and 3E). Importantly, we demonstrated that α-syn fibrils trigger LMP also in the recipient cells following their intercellular transfer (Fig 8A and 8B). LMP in acceptor cells has been only reported in a coculture system where donor N27 cells expressing α-syn were pretreated with the mitochondrial toxin MPP⁺ (1-methyl-4-phenylpyridine) to induce PD-like pathology and cocultured with acceptor cells stably expressing Gal3 [73]. Our results indicate that, without inducing a PD-like pathology in donor cells, the transfer of α-syn fibrils "per se" is sufficient to induce LMP in acceptor cells both in homotypic (Fig 8A and 8B) and heterotypic coculture systems (Fig 10F and 10G). In addition, by performing SR microscopy, we also detected the recruitment of Gal3 to damaged lysosomes bearing α-syn fibrils either in their lumen or at their membrane in the acceptor cells, similar to what we observed in donor cells (Fig 8C and 8D). Interestingly, our results show that lysosomes containing α-syn fibrils are not targeted for lysophagy [84] (S4 Fig), possibly suggesting a rescue mechanism such as the one orchestrated by the endosomal sorting complexes required for transport (ESCRT) machinery [111,112]. Considering the evidence supporting autophagy impairment in NDs and more specifically following α-syn accumulation [55,113], and the recent findings that impaired autophagic flux contributes to the occurrence of LMP [114], it will be interesting to assess the contribution of autophagy to LMP, lysosomal dysfunction, and propagation of α-syn.

The underlying mechanism of LMP induction by α-syn fibrils is unclear. Several studies have reported the ability of α-syn protofibrils to permeabilize membranes [115–117]; thus, as suggested by Freeman and colleagues, the luminal α-syn fibrils could induce membrane curvature causing the rupture of lysosomes [73]. Our correlative EM data show a wavy limiting membrane in lysosomes containing α-syn fibrils compared to lysosomes devoid of α-syn fibrils, supporting this hypothesis. Moreover, oxidative stress could also compromise lysosomal integrity [118]. Consistently, we have previously reported that α-syn fibrils increase reactive oxygen species (ROS) levels in CAD cells [19], and enhanced ROS levels have been found in PD mouse models and PD patients [68,119].

Since α-syn fibrils located to the lysosomes need to directly interact with the soluble cytosolic α-syn in order to induce the seeding, LMP has been proposed as a relevant mechanism to allow the escape of α-syn fibrils from the lysosomes [19,85]. Colocalization between Gal3 and

aggregated α-syn has been reported in the basal forebrain of diffuse Lewy body disease patients but not in normal, age-matched controls, supporting the role of lysosome rupture in α-syn pathology and progression [85]. In the current study, we detected efficient seeding of soluble α-syn both in donor cells treated with α-syn fibrils and in acceptor cells that received the fibrils inside lysosomes from donor cells (Figs 4A, 4B, 9A and 9B). By using SR microscopy, we observed that 69 ± 6% and 60 ± 7% of the newly formed α-syn aggregates colocalize with LAMP1+ lysosomes in the donor and acceptor cells, respectively. These data, together with live imaging, show that the seeding event begins shortly after the internalization of α-syn fibrils and occurs preferentially at the lysosomes (Figs 4C–4I and 9C–9G, S5A Fig, S1 Movie). It has been shown that specific environments inside membranous and vesicular structures favor the aggregation of the α-syn protein [120,121]; thus, our results indicate that lysosomes can create a confined environment facilitating the conversion of soluble α-syn.

Do lysosomes contribute to the spread of the disease? Lysosomes are highly dynamic organelles that move bidirectionally along microtubules between the center and the periphery of the cell [122]. In nonpolarized cells, lysosomes are mostly clustered at the perinuclear region, but a more dynamic pool is located at the cell periphery [90,92,100]. Here, we demonstrated that α-syn fibrils act on lysosome positioning by inducing their peripheral redistribution (Fig 5A and 5B). We also found that peripheral α-syn–containing lysosomes exhibit lysosomal dysfunction (i.e., disturbed pH and decreased lysosomal enzyme activities; Fig 5C). In addition, we showed that donor cells challenged with α-syn fibrils, thus having more peripheral lysosomes, transfer lysosomes more efficiently to the recipient cells compared to untreated donor cells (Fig 5F and 5G). Moreover, the peripheral distribution of lysosomes positively affects α-syn fibrils' transfer both in CAD and HeLa cells (Fig 7A and 7B, S6A and S6B Fig).

The induction of LMP has been recently shown to promote nuclear translocation of TFEB [123], which influences lysosomal motility and exocytosis [95]. Furthermore, TFEB overexpression promotes α-syn clearance [124]. Thus, we investigated whether the loading of lysosomes with α-syn fibrils triggers TFEB nuclear translocation and whether this translocation participates in α-syn–induced distribution of lysosomes to the cell periphery. We found that α-syn fibrils induced TFEB translocation to the nucleus (Fig 6A). Moreover, expression of active TFEB correlates with more peripheral distribution of lysosomes (Fig 6B), while silencing of the endogenous TFEB prevents the peripheral redistribution of lysosomes upon α-syn loading (Fig 6D), suggesting the involvement of TFEB in this event. However, how this occurs is not yet clear. Previous reports suggested that TFEB-regulated transcription induced lysosomal docking to the PM, thus promoting the peripheral distribution of lysosomes in HeLa cells [95]. However, TFEB and the related transcription factor TFE3 have been reported to contribute to perinuclear positioning of lysosomes through transcriptional activation of the lysosomal protein TMEM55B under starvation conditions in HeLa cells [96]. It has also been reported that overexpression of TFEB promotes perinuclear positioning of the lysosomes in mouse fibroblast cells defective for the lysosomal transporter cystinosin [97]. Given these controversies in the literature, the exact role of TFEB in the regulation of lysosome positioning remains to be addressed in the future.

To our knowledge, this is the first report showing that α-syn fibrils influence lysosome positioning, favoring α-syn fibrils transfer in a cell-to-cell contact-dependent manner via lysosomes. These findings suggest that lysosomes play an important role in the intercellular trafficking of α-syn fibrils. It is worth noting that α-syn fibrils do not affect lysosomal exocytosis (Fig 5D and 5E). In addition, in the secretion tests used in each coculture experiment, we never detected any significant difference in the transfer of α-syn fibrils and/or lysosomes through the cell supernatant. Although this was not the aim of our study, these results further confirm that transfer of α-syn fibrils primarily relies on TNT- and/or cell-to-cell-contact–dependent mechanisms in

accordance with our previous findings [19,20,22]. The mechanism by which α-syn fibrils enhance TNT formation is still largely unknown. However, we and others demonstrated that cells form more TNTs under stress, such as when they are treated with $H_2O_2$ [125,126]. In line with these findings, we previously reported that CAD cells bearing α-syn fibrils exhibit sustained increase in ROS levels [19]. Therefore, α-syn fibrils can indeed trigger stress in the cells in order to induce TNT formation. In addition, it has been reported that α-syn could induce membrane remodeling, such as membrane expansion and formation of cylindrical tubes, and that the extent of membrane expansion correlated linearly with the density of α-syn [127]. Thus, further studies are essential to address how α-syn fibrils participate in the formation of TNTs.

Lysosome transfer through TNTs has been shown to be bidirectional [106]. However, there is no study on the directionality/origin and the nature (whether they are healthy or damaged) of the lysosomes transferred between cells in the presence of the α-syn fibrils. Thus, in the current study, we studied the transfer of both healthy and damaged (i.e., under LMP) lysosomes between CAD (donor) and HeLa (acceptor) cells and found that the transfer of both types of lysosomes is bidirectional (Fig 10A, 10B, 10H and 10I). We showed that α-syn fibril-treated donor cells transfer more lysosomes (both healthy and damaged) to acceptor cells compared to untreated control donor cells. Interestingly, we observed the recruitment of acceptor's Gal3 to the donor's lysosomes transferred in an acceptor cell (S7G Fig), suggesting that the rupture of donors' lysosomes containing α-syn fibrils could occur after their transfer in the recipient cells. Focusing on the transfer of lysosomes from acceptors to donors, we found that α-syn fibril-treated donor cells received more healthy acceptors' lysosomes compared to control untreated donor cells (Fig 10H and 10I). We also detected damaged acceptors' lysosomes in α-syn fibril-treated donor cells, although this was a very rare event (S7H and S7I Fig). Based on these data, we suggest that in the context of α-syn spreading, donor cells loaded with α-syn fibrils try to get rid of α-syn fibrils through the transfer of lysosomes, thus promoting the spreading of the fibrils. On the other hand, donor cells receive healthy lysosomes from acceptor cells, suggesting a potential rescue mechanism. A similar rescue mechanism has been previously reported in cystinosin-deficient fibroblasts, which receive cystinosin-bearing lysosomes from healthy macrophages through TNTs [106].

In light of these findings, we propose a working model on how lysosomes contribute to the transfer and propagation of α-syn fibrils (Fig 11). After internalization, α-syn fibrils localize to lysosomes (Fig 11, number 1), where they induce LMP and recruit Gal3 (Fig 11, number 2). α-Syn fibrils escape from damaged lysosomes and seed soluble α-syn in donor cells, an event that occurs mostly in association with the lysosomes (Fig 11, number 3). Following α-syn fibrils internalization, TFEB translocates to the nucleus (Fig 11, number 4) and participates in the redistribution of lysosomes containing α-syn fibrils, likely under LMP or functionally impaired, toward the cell periphery (Fig 11, number 5). Here, lysosomes are not exocytosed (Fig 11, number 6), but are more prone to be transferred to other cells through TNTs. Indeed, cells loaded with α-syn fibrils transfer more lysosomes (both under LMP or not) to acceptor cells compared to healthy cells through TNTs (Fig 11, number 7). Once lysosomes reach the acceptor cells, α-syn fibrils escape from lysosomes already under LMP or from lysosomes that undergo LMP when they arrive in the acceptor cells. This again allows α-syn fibrils to seed soluble α-syn in the acceptor cells, mostly in association with the lysosomes (Fig 11, number 8) and further triggering LMP (Fig 11, number 9). In an attempt to rescue α-syn-overloaded donor cells, acceptor cells transfer healthy lysosomes to donor cells (Fig 11, number 10).

In conclusion, our data explain how lysosomes are involved in the transfer and propagation of α-syn aggregates and the crucial role that they play in this process. In agreement with these findings, we have previously reported that α-syn fibrils can be efficiently transferred between

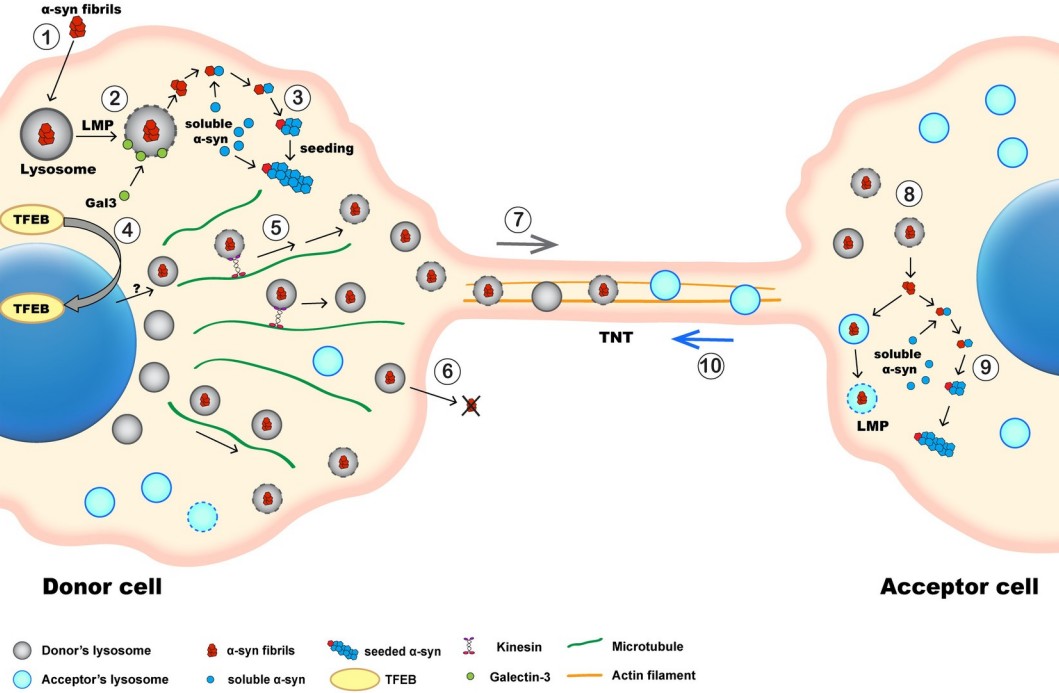

**Fig 11. Working model. (1)** α-Syn fibrils localize to lysosomes following their uptake. **(2)** α-Syn fibrils induce LMP (i.e., recruit Gal3 to the ruptured lysosomes) in donor cells following their uptake, allowing their escape from lysosomes. **(3)** α-Syn fibrils interact with soluble α-syn and induce seeding in donor cells. **(4)** Following α-syn fibrils internalization, TFEB translocates to the nucleus. **(5)** Lysosomes (possibly under LMP, as their pH is higher and lysosomal activities is lower) having the α-syn fibrils move toward the cell periphery. **(6)** Peripheral lysosomes are not preferentially exocytosed. **(7)** Lysosomes having α-syn (both under LMP or not) are transferred to acceptor cells along with healthy lysosomes. **(8)** α-Syn fibrils induce LMP following their transfer to acceptor cells. **(9)** α-Syn fibrils induce further seeding following their transfer to acceptor cells. **(10)** Acceptor cells transfer healthy lysosomes to donors. α-syn, α-synuclein; Gal3, Galectin-3; LMP, lysosomal membrane permeabilization; TFEB, transcription factor EB; TNT, tunneling nanotube.

neurons, primary astrocytes, neurons and astrocytes, astrocytes and organotypic hippocampal slices and between iPSCs, and they were associated with lysosomes in these coculture systems [19,21,56]; however, we have used here undifferentiated neuronal CAD cells line, for their easy manipulation. Although our results contribute to a better understanding of the mechanism of development and progression of α-syn pathology, opening new avenues for possible therapeutic approaches against synucleinopathies, it will be necessary in the future to investigate the transfer of α-syn containing lysosomes in differentiated primary neurons and/or in ex vivo/in vivo systems to further validate our data and overcome the limitation of this study.

## Materials and methods

### Cell culture, transfection, and siRNA treatment

The mouse catecholaminergic neuronal cells (CAD cells), generated from a brain tumor of transgenic mice by targeted oncogenesis (kind gift of Hubert Laude; Institut National de la Recherche Agronomique, Jouy-en-Josas, France), were cultured in Opti-MEM GlutaMAX (Gibco, Thermo Fisher Scientific, Waltham, Massachusetts, USA) including 10% fetal bovine serum (FBS) and 1% penicillin-streptomycin (Pen-Strep). CAD cells express neuronal properties, but they lack neuronal morphology, thus stay undifferentiated in the presence of serum in their medium. This situation can be reversed by serum starvation, and differentiation can be induced [128]. In this study, we used CAD cells in their undifferentiated state as this condition

is optimal to study TNTs [19,125,129–131]. HeLa cells WT, HeLa cells myrlysin-KO and KIF1B-KIF5B double KO [100,132], and HeLa cells stably expressing Gal3-Turquoise were cultured in DMEM-GlutaMAX including 4.5 g/L D glutamine and pyruvate (Gibco) supplemented with 10% of FBS. CAD and HeLa cells were transiently transfected with Gal3-GFP, Arl8b-GFP, fluorescently tagged human histone H2B (H2B-GFP or H2B-mCherry) plasmids (final concentration: 1 μg for 150,000 cells/35mm Ibidi μ-dishes, 3 μg for 800,000 cells/T25 flasks) by Lipofectamine 2000 (Invitrogen, Carlsbad, California, USA) according to the manufacturer's instructions.

To down-regulate Arl8b and TFEB, 30 pmol of unique 27mer siRNA duplex against mouse Arl8b and TFEB (CliniSciences, Nanterre, France and OriGene, Rockville, Maryland, USA, respectively) was introduced to CAD cells in parallel to siRNA scramble (sicontrol) by using Lipofectamine RNAiMAX (Invitrogen) according to the manufacturer's instructions and incubated 60 hours prior to be used in coculture experiments and in western blot analysis.

To induce lysophagy, CAD cells were treated with 1 mM LLOMe for 3 hours.

## Generation of stable HeLa mTurquoise-Galectin3 cell line

In order to generate stable HeLa cell line expressing a mTurquoise-Galectin3 fluorescent reporter, we used a well-established Sleeping Beauty (SB) transposable system that originates from an inactive transposon in salmonid fish and belongs to the Tc1/mariner-type transposon family, as previously described [133,134]. Briefly, the insert sfiI-mTurquoise-Galectin3-sfiI was amplified from pE mTurquoise-Galectin3 plasmid by polymerase chain reaction (PCR) where SfiI-mTurquoise-forward (AGG CCT CTG AGG CCG CCA CCA TGG TGA GCA AGG G) and Gal3-SfiI-reverse (AGG CCT GAC AGG CCT TAT CAT GGT ATA TGA AGC AC) primers (Integrated DNA Technologies, Coralville, Iowa, USA) were used with Flash Phusion High Fidelity PCR Master Mix (Thermo Fisher Scientific, Eindhoven, the Netherlands). Next, the insert was cut by SfiI enzyme and inserted into the constitutive vector pSBbi-PUR, a gift from Eric Kowarz (Addgene Plasmid #60523), which was cut by sfiI enzyme and dephosphorylated by Rapid DNA Dephos & Ligation Kit (Roche, Basel, Switzerland). Transformation was performed with on stBL3 competent bacteria following the ligation of sfiI-pSBbi-PUR-sfiI vector and insert sfiI-mTurquoise-Galectin3-sfiI, and colonies were selected on LB broth agar plates containing amphicilin. Clones were then sequenced (Eurofins Genomics, Luxembourg) in order to verify the existence of the insert, and maxiprep for the pSBbi-PUR-mTurquoise-Galectin3 plasmid was performed by using the Invitrogen PureLink HiPure Plasmid Maxiprep Kit. Next, HeLa cells seeded on 6-well plates (300,000 cells per well) were co-transfected with 9 μg of pSBbi-PUR-mTurquoise-Galectin3 and 100 ng SB transposase expression vector pCMV (CAT)T7-SB100, a gift from Eric Kowarz (Addgene plasmid # 34879) by using Xtrem-Gene transfection reagent (Roche). Moreover, 24 hours following the transfection, cells were subjected to puromycin (1 μg/mL). Selection was carried out for 2 to 10 days. Cells were then cultivated for 4 weeks without selection medium, and monoclonal selection was performed on 96-well plates (1 cell/well). Clones exhibited expected mTurquoise fluorescent were further tested for classical behavior for HeLa cells (low Gal3 fluorescent in the whole cell and a brilliant signal upon vacuole rupture) and used in the experiments described in the current study.

## Preparation and internalization of α-syn fibrils

Human WT α-syn in pRK172, a construct containing α-syn that lacks cysteine because of mutagenesis of codon 136 (TAC to TAT), was transformed into *Escherichia coli* BL21 (DE3) as previously reported. Expression and purification were performed as described previously [86]. Briefly, the protein concentrations of monomeric α-syn were measured by RP-HPLC as

described previously [86,135]. Purified recombinant α-syn monomers (approximately 5 mg/mL) containing 30 mM Tris-HCl, pH 7.5, 10 mM DTT, and 0.1% sodium azide were incubated at 37°C with shaking using a horizontal shaker (TAITEC) at 200 rpm. Following the incubation for 7 days, the samples were ultracentrifuged at 100,000 g for 20 minutes at room temperature (RT), and the ppt fraction was recovered as α-syn fibrils. They were resuspended in saline prior to be ultracentrifuged again. The pellets were resuspended in saline and sonicated using a cup horn sonicator (Sonifier SFX, Branson, Brookfield, Connecticut, USA) at 35% power for 180 seconds (total 240 seconds, 30 seconds on, 10 seconds off). The fibrils were labeled with Alexa Fluor 488 or 568 Protein Labelling Kit (Invitrogen) following the manufacturer's instructions. After the incubation of fibrils with Alexa Fluor dyes, samples were ultracentrifuged again, and the pellets were resuspended in 30 mM Tris-HCl, pH 7.5 prior to be ultracentrifuged one more time. Finally, labeled α-syn fibrils were resuspended in saline containing 0.1% sodium azide, and the protein concentration of the fibrils was determined by RP-HPLC as described before [86,135]. Seeding activity of the fibrils were checked by adding the labeled fibrils (3 μg) to 100 μL of 1 mg/mL α-syn monomer solution in 30 μM Thioflavin T and 80 mM Hepes, pH 7.5. Amyloid-like fibril formation was continuously monitored by measuring the Thioflavin T fluorescence (excitation 442 nm, emission 485 nm) with a microplate reader (Infinite 200, TECAN).

A total of 0.5 μM Alexa 488 or Alexa 568–tagged α-syn fibrils were diluted in the appropriate medium and sonicated for 5 minutes at 80% amplitude with a pulse cycle of 5 seconds on and 2 seconds off in an ultrasonic water bath Vibra-Cell 75,041 (BioBlock Scientific, Strasbourg, France). Fibrils were then introduced to cells immediately and incubated for 18 hours. Before cells were processed for further procedures (e.g., coculture preparation, fixation, etc.), they were washed 3 times with diluted trypsin (1/3) in Phosphate-buffered saline (PBS). In parallel, control cells were subjected to the same treatments as described above without the addition of α-syn fibrils.

## Immunocytochemistry for lysosomes

CAD and/or HeLa cells and cocultures (CAD-CAD and CAD-HeLa) were fixed with 4% paraformaldehyde (PFA) for 20 minutes at RT. After 3 washes with PBS, cells were permeabilized with 0.1% Triton X-100 in PBS for 3 minutes at RT. Cells were then washed with PBS, and nonspecific antibody binding sites were blocked by using 10% of goat serum (GS) in PBS for 30 minutes. Cells were then incubated with primary rat monoclonal anti-mouse LAMP1 (1D4B) or anti-human LAMP1 (H4A3) primary antibodies with the dilution of 1/200 in blocking solution for 1 hour. Antibodies were purchased from Developmental Studies Hybridoma Bank (DSHB, Iowa City, Iowa, USA). Cells were then washed 3 times with PBS and incubated for secondary antibodies (anti-rat Alexa-647 or anti-rat Alexa-488 purchased from Invitrogen) diluted 1:500 in blocking solution for 2 hours at RT. After 3 washes with PBS, cells were further stained with HCS CellMask Blue (Invitrogen) for 15 minutes at RT (diluted 1:5,000 in PBS). Cells were then washed 3 times with PBS and further stained for DAPI (diluted 1:5,000 in PBS) for 5 minutes. Finally, cells were washed and mounted with Aqua-Poly/Mount (Polysciences, Warrington, Pennsylvania, USA). Image acquisitions were performed by confocal microscopy with identical settings.

## Immunocytochemistry for lysophagy

CAD cells were fixed with 4% PFA for 20 minutes at RT. After 3 washes with PBS, cells were permeabilized with 0.1% Triton X-100 in PBS for 3 minutes at RT. Cells were then washed with PBS, and nonspecific antibody binding sites were blocked by using 2% of bovine serum

albumin (BSA) in PBS for 30 minutes. Cells were then incubated with primary rabbit anti-mouse LC3 (1:500, MBL International, Woburn, Massachusetts, USA), mouse anti-Ubiquitin (1:1,000, FK2, Enzo, Farmingdale, New York, USA), and rat monoclonal anti-mouse LAMP1 (1D4B) (1:100, DSHB) in blocking solution for 1 hour. Cells were then washed 3 times with PBS and incubated for secondary antibodies (anti-rabbit Alexa 488, anti-mouse Alexa-647 from Invitrogen; and anti-rat CF405M from Sigma, Saint Louis, Missouri, USA) diluted 1:500 in blocking solution for 1 hour at RT. Finally, cells were washed and mounted with Aqua-Poly/Mount (Polysciences). Image acquisitions were performed by confocal microscopy with identical settings.

## LysoTracker Deep Red staining, DQ-BSA assay, and Cathepsin B activity assay

CAD cells were plated on 35-mm Ibidi μ-dishes (100,000 cells/dish). Alexa 568–tagged α-syn fibrils were introduced or not (control) to the cell culture medium 24 hours after plating. Following 18 hours of α-syn fibril uptake, 20 nM LysoTracker DR (Invitrogen) was prepared in culture medium, and cells were incubated for 20 minutes at 37˚C. Cells were then fixed with 4% PFA at RT and stained for HCS CellMask Blue and DAPI as described before. Image acquisitions were performed by confocal microscopy with identical settings.

To detect CathB activity, CAD cells were plated on 35-mm Ibidi μ-dishes (100,000 cells/dish). Alexa 488-tagged α-syn fibrils were introduced or not (control) to the cell culture medium 24 hours after plating. Following 18 hours of α-syn fibril uptake, cells were treated with 1X Magic Red Cathepsin solution, provided by Magic Red Cathepsin Assay kit (Immuno-Chemistry Technologies, Bloomington, Minnesota, USA) and prepared through the manufacturer's recommendation, for 15 minutes at 37˚C. Cells were then immediately imaged by confocal microscopy in live condition. The intracellular activity of CathB was detected as red fluorescent puncta produced by the hydrolysis of the fluorogenic substrate introduced. All images were acquired with the identical settings. For both experiments, integrated density of LysoTracker DR and CathB signal was measured by using Fiji software after a region of interest (ROI) is manually created by the freehand selection tool and the measurements were expressed in AU.

For DQ-BSA assay, CAD cells were plated on 35-mm Ibidi μ-dishes (130,000 cells/dish). Alexa 568–tagged α-syn fibrils were introduced or not (control) to the cell culture medium 24 hours after. Following 18 hours of α-syn fibrils uptake, 10 μg/mL DQ Green BSA (Thermo Fisher Scientific) was prepared in culture medium, and cells were incubated for 90 minutes at 37˚C. Cells were then fixed with 4% PFA at RT. Image acquisitions were performed by confocal microscopy with identical settings. Integrated density of DQ-BSA signal was measured by using Fiji software after a ROI is manually created by the freehand selection tool, and the measurements were expressed in AU.

## TNT labeling

To preserve the structure of TNTs, cells were fixed by a 2-step fixation protocol as described previously [20,129]. Briefly, cells were first fixed with fixative solution-1 (2% PFA, 0.05% glutaraldehyde (GA) and 0.2 M HEPES in PBS) for 20 minutes at 37˚C, then with fixative solution-2 (4% PFA and 0.2 M HEPES in PBS) for an additional 20 minutes at 37˚C. After several washes, cells were labeled with 3.3 μg/μL WGA-Alexa Fluor-488 nm conjugate solution (Life Technologies, Thermo Fisher Scientific, Waltham, Massachusetts, USA) for 20 minutes at RT for membrane detection. As there is no specific marker for TNTs, WGA labeling was used to detect TNTs as described in detail in the quantification section.

## Lysosomal exocytosis analysis

**Evaluation of released Cathepsin B by ELISA.** CAD cells were plated in B12-wells (80,000 cells/well). Cells were then treated with Alexa 568–tagged α-syn fibrils or not for 18 hours. Soluble lysosomal CathB released in culture medium was evaluated by using Human Cathepsin B ELISA Kit (Abcam, Cambridge, UK) following the manufacturer's instructions. Briefly, aliquots of culture medium derived from control cells and cells challenged with α-syn fibrils were incubated in 96-well plates precoated with a CathB-specific mouse monoclonal antibody for 90 minutes at 37˚C. Biotinylated CathB-specific goat polyclonal antibody was then added on the cells for 60 minutes at 37˚C. Cells were then washed with Tris-buffered saline (TBS) prior to the addition of Avidin-Biotin-Peroxidase Complex for 30 minutes at 37˚C. Unbound conjugates were washed away with TBS buffer 5 times. Moreover, 3,3′,5,5′-tetramethylbenzidine was then used to visualize the horseradish peroxidase (HRP) enzymatic reaction (O.D. absorbance: 450 nm). Released CathB content is expressed as nanograms of CathB/mL of culture medium.

**Evaluation of LAMP1 at the plasma membrane.** CAD cells were plated in 35-mm Ibidi μ-dishes (120,000 cells/dish). Cells were then treated with Alexa 568–tagged α-syn fibrils or not for 18 hours. Live cells were the incubated with the primary rat monoclonal anti-mouse Lamp1 (1D4B) antibody recognizing an epitope at the luminal region of the protein, diluted in complete medium, for 30 minutes at 4˚C. After washing with PBS, cells were fixed in 2% PFA in PBS for 10 minutes at RT. Next, cells were incubated with the Alexa Fluor 488–conjugated goat anti-rat secondary antibody diluted in PBS for 30 minutes at RT. Cells were then imaged by confocal microscopy.

## Detection of LMP and seeding

CAD cells were plated on 35-mm Ibidi μ-dishes (100,000 cells/dish) and transiently transfected with 1 μg of Gal3-GFP (for LMP) or 1 μg of α-syn-GFP plasmids (for seeding). Moreover, 24 hours later, cells were either treated with Alexa 568–tagged α-syn fibrils for 18 hours or left untreated. In coculture experiments, cells transiently transfected with Gal3-GFP or α-syn-GFP plasmids were cocultured for 24 hours with donor cells previously treated for 18 hours with Alexa 568–tagged α-syn fibrils or not. Cells were then fixed with 4% PFA for 20 minutes at RT and stained for DAPI. Following image acquisition with confocal microscopy, cells were quantified for the presence of Gal3-GFP or α-syn-GFP puncta by using "Spot detector" tool of the ICY software. Note that a threshold of 5 puncta is applied to all images analyzed, and cells bearing puncta above this threshold were considered as having LMP.

## Coculture experiments

Donor cells were plated on 35-mm well plates (400,000/well) and transfected with suitable plasmids according to the experiment (LAMP1-GFP, Arl8b-GFP, Gal3-GFP, and H2B-mCherry) or pretreated with siRNA Arl8b or siRNA-scrambled for 60 hours. Myrlysin-KO and KIF1B-5B dKO HeLa cells were directly used as donors. Donor cells (in parallel to untreated control cells) were then challenged with Alexa 568–tagged α-syn fibrils for 18 hours (or not according to the experiment). They were then washed once with PBS and 3 times with 1:3 diluted trypsin before coculture preparation.

Acceptor cells were plated on T25 flasks (800,000/flask) and transfected with suitable plasmids according to the experiment (Gal3-GFP, H2B-mCherry, H2B-GFP, and α-syn-GFP). HeLa Gal3-Turquoise cell line was used directly as acceptor cell population. Both donor and acceptor cells were then detached (mechanically and by trypsinization for CAD and HeLa cells, respectively) and counted. Coculture was prepared in a 1:1 ratio (100,000 donor and

100,000 acceptors on 35-mm Ibidi μ-dishes) for 24 hours. Cells were then fixed with 4% PFA for 20 minutes at RT and stained for HCS CellMask Blue and DAPI as described before and mounted. Schematic presentation of different coculture preparations was presented in S3 Fig.

Of note, cocultures were named with the predominant treatment of the donor cells: (i) cocultures having donor cells treated with α-syn fibrils or not were referred to as "α-syn coculture" and "control coculture," respectively; and (ii) in the cases where all donor populations were treated with α-syn fibrils, in order to emphasize the different treatments between donor cells, cocultures were named by these treatments (e.g., Arl8b-GFP over-expressing or siArl8b pretreated donors were referred to as "Arl8b-GFP coculture" and "siArl8b coculture," respectively).

**Secretion test.** In all coculture conditions, donor cells and acceptor cells were also plated separately during the coculture preparation (100,000 cells/35-mm Ibidi μ-dishes) for the total duration of coculture incubation (24 hours). Then the conditioned medium collected from donor cells were transferred onto acceptor cells. Acceptor cells were then quantified for the presence of specific cargos (e.g., lysosomes and α-syn fibrils) to assess the cargo transfer through secretion in each coculture condition. Secretion tests for particular experiments were referred to as predominant treatment of the donor cells as described above (e.g., "control secretion," "α-syn secretion," and "Arl8b-GFP secretion").

## Confocal microscopy

After fixation and immunostaining, images were acquired with an inverted laser scanning confocal microscope LSM700 (Zeiss, Germany), with a 63X objective (zoom 1.0). Images were acquired using the ZEN acquisition software (Zeiss). In all experiments, we acquired Z-stacks covering the whole volume of cells. For the experiments where integrated density was measured in control and α-syn treated cells (LysoTracker DR, Magic Red CathB, DQ-BSA assays, and evaluation of LAMP1 at the PM in non-permeabilized condition), all settings (including the laser power and exposure line time) were kept identical. In some experiments, line averaging were also used in order to improve the signal-to-noise ratio.

## Super-resolution microscopy

**Sample preparation.** The 24-well plates containing sterile coverslips (Deckglaser 12 mm) were coated with fibronectin (25 μg/mL) diluted in PBS and kept at 37°C for 1 hour. Coverslips were then washed 3 times with PBS. CAD cells were then plated on coverslips (20,000 cells/coverslip). Gal3-GFP or α-syn-GFP transfected and α-syn treated cells or mixture of donor and acceptor cells (cocultures prepared for LMP and seeding experiments) were then plated on coverslips. Cells were then fixed with 4% PFA for 20 minutes at RT and immunolabeled with primary rat monoclonal anti-mouse LAMP1 (1D4B) and secondary anti-rat Alexa 647 antibodies as described before. Cells were then stained with DAPI and HCS CellMask Blue and mounted with Fluoromount-G (Southern Biotech, Birmingham, Alabama, USA) mounting medium.

**Structured illumination microscopy.** SIM was performed on a Zeiss LSM780 Elyra PS1 microscope (Zeiss) using 100X/1.4 oil Plan Apo objective with a 1.518 refractive index oil (GE Healthcare Life Sciences, USA) and an EMCCD Andor Ixon 887 1 K camera for the detection. SIM images were processed with ZEN software and then aligned with ZEN using 100-nm TetraSpeck microspheres (Thermo Fisher Scientific) embedded in the same conditions as the sample.

**Spinning disk live super-resolution confocal microscopy.** Images were acquired with an inverted Eclipse Ti Nikon microscope equipped with a CSU-X1 spinning disk confocal

scanning unit (Yokogawa, Tokyo, Japan), with an EMCCD Camera (Evolve 512 Delta, Photometrics, USA), and with Live SR super-resolution module (Gataca Systems, Massy, France), using a ×100 or ×60 1.4 NA PL-APO VC oil objective lenses controlled by MetaMorph software. SR images were denoised by using Safir software [136].

### Live imaging microscopy

Time-lapse microscopy imaging was performed on an inverted spinning disk microscope (Elipse Ti microscope system, Nikon Instruments, Melville, New York, USA) using 60X 1.4 NA CSU oil immersion objective lens and laser illumination 488 only or together with 561. For live cell imaging, the 37˚C temperature was controlled with an Air Stream Stage Incubator, which also controlled humidity. Cells were plated in Ibidi μ-Dish 35 mm and incubated with 5% $CO_2$ during image acquisition. Image processing and movies were realized using Meta-Morph and ImageJ/Fiji software.

### Electron microscopy

**Correlative resin electron microscopy.** For correlation of FM and resin-embedded EM of CAD cells, FM imaging was performed prior to sample preparation in EM. Cells were transiently transfected with LAMP1-GFP using Lipofectamine 2000 transfection reagent for 24 hours and seeded on carbon-coated, gridded coverslips prepared as described before [76]. Moreover, 24 hours later, they were treated with α-syn Alexa 568 fibrils for 18 hours. Cells were then washed in 0.1 M phosphate buffer (PB) and fixed in 4% formaldehyde in 0.1 M PB buffer after which fluorescent Z-stacks of cells of interest were obtained for the GFP and Alexa 568 signal. The position of cells relative to the pattern etched in the coverslip was registered using polarized light. After FM imaging, the same specimens were prepared for EM by fixing with 2.5% GA + 2% formaldehyde in 0.1 M PB buffer for 2 hours, and postfixation with osmium tetroxide and uranyl acetate. Fixed cells were dehydrated using a graded ethanol series and embedded in Epon resin, which was polymerized for 48 hours at 65˚C. After polymerization, the glass coverslip was removed from the Epon block by dissolving it in hydrogen fluoride. The exposed Epon surface was thoroughly cleaned with distilled water and left to harden overnight at 63˚C. Areas of the resin block containing the cells previously imaged by FM were cut out using a clean razor blade and glued to empty Epon sample stubs, with the basal side of the cells facing outwards [75,76]. From these blocks, 70-nm sections were cut and collected on formvar and carbon coated copper 50 mesh support grids. Thin EPON sections were imaged in a Tecnai 12 transmission electron microscopy (TEM; Thermo Fisher Scientific) equipped with a Veleta 2k×2k CCD camera (EMSIS, Munster, Germany), operating at 80 kV. For correlation of FM and EM data, the TEM image of a ROI was overlaid with FM data using Adobe Photoshop. Multiple corresponding spots (e.g., nuclei) on images were manually selected, after which the correct scaling and transformation steps were performed, and high-precision overlays of FM and EM data were generated. We used only linear transformation options to achieve the overlays shown in the figures.

**On-section correlative microscopy of thawed cryosections.** Similarly, CAD cells were grown on carbon-coated, gridded coverslips, incubated with α-syn Alexa 568–tagged fibrils for 18 hours and fixed in 0.1 M PB (pH 7.4) containing 4% PFA that was added to an equal volume of medium. Fixed cells were flat embedded in 12% gelatin by keeping their orientation, cryoprotected in 2.3 M sucrose, and plunge frozen according to previous protocols [137,138]. Moreover, 70-nm cryosections were picked up on formvar-coated copper grids, after which they were immunolabeled for LAMP1 with a secondary Alexa 488-tagged rabbit antibody and 10-nm gold-conjugated Protein A (Cell Microscopy Core, UMC Utrecht, the Netherlands).

Finally, sections were labeled using Hoechst 33342 to outline nuclei for correlation. The grids were washed with dH2O and sandwiched between a microscope slide and a no. 1 coverslip in 2% methylcellulose in $dH_2O$. Sections were imaged in a Deltavision RT wide field fluorescence microscope (GE Healthcare Life Sciences, USA) equipped with a Cascade II EM-CCD camera (Photometrics). Grids were first imaged at 40× magnification to form a map of the section, after which ROIs were selected using 100× magnification. After imaging, the grids were removed from the microscope slide, thoroughly rinsed with $H_2O$ and contrasted for EM, and embedded in methylcellulose containing uranyl acetate. After drying, ROIs were imaged using a Tecnai 12 TEM (Thermo Fisher Scientific) operating at 80 kV, equipped with a Veleta 2k×2k CCD camera (EMSIS) camera, and running serialEM software. Registration of thin section FM and EM data was performed using Adobe Photoshop, similar to above.

## Western immunoblots

For the western blot analysis, cells were rinsed with PBS and then scraped in RIPA buffer (50 mM Tris–HCl, pH 7.5, 150 mM NaCl, 1% Triton X-100, 0.5% sodium deoxycholate, 0.1% SDS). The resulting soluble fractions were centrifuged at 18,000 g for 5 minutes at 4˚C, and then, the supernatants were taken as the whole cell lysate, mixed with Laemmli sample buffer (1 M Tris–HCl, pH 6.8, 20% glycerol, 4% SDS, 5% ß-mercaptoethanol, and 0.02% bromophenol blue), and boiled for 5 minutes. For western blots, equal amounts of proteins (30 μg/lane) were separated on SDS-PAGE gels and transferred to nitrocellulose membranes (GE Healthcare Life Sciences), following standard procedures. Membranes were then blocked in TBS containing 0.1% Tween-20 and 5% nonfat dried milk for 1 hour and probed overnight at 4˚C with primary antibodies. The membranes were then washed and exposed for 1 hour at RT to the anti-rabbit/anti-mouse peroxidase-conjugated antibodies (1:1,000). The specific protein bands were visualized using the ECL immunoblotting chemiluminescence system (GE Healthcare Life Sciences) and the ImageQuant LAS 500TM camera (GE Healthcare Life Sciences). Primary antibodies used in this study are the following: rat anti-Lamp1 (1:100, DSHB), anti-Arl8b (1:100), rabbit anti-TFEB (1:1,000, Bethyl Laboratories, Montgomery, Texas, USA), mouse anti-tubulin (1:1,000), and rabbit anti-GAPDH (1:5,000, Boster, Pleasanton, California, USA). To determine the apparent molecular weights of the protein bands, a PageRuler plus prestained protein ladder (Thermo Fisher Scientific) was used.

## Quantification analysis

**Quantification for lysosomal positioning.** We performed 2D analysis with the Z-projection where we used a 2-step detection protocol in ICY software in order to assess the lysosomal positioning. In the first step, we used a script to detect 2 ROIs: (i) the limiting membrane of whole cell that was detected with HCS CellMask Blue staining; and (ii) the nuclear region that was detected by DAPI staining. It is important to note that in the cases where perinuclear region of the cells including microtubule organizing center (MTOC) was not detected in Z-projection, this ROI was manually corrected. In the second step of the protocol, another script was used, where "Spot detector" and "Colocalization" modules were combined. Thus, we were able to detect the number of lysosomes, the number of α-syn fibrils puncta, and the number of lysosomes colocalizing with the α-syn fibrils. Number of peripheral lysosomes were calculated by simply subtracting the number of centrally located lysosomes from the whole number of lysosomes, and the percentages central and peripheral lysosomes were calculated accordingly. Same protocols were used to quantify the LysoTracker DR and MagicRed Cath B positive lysosomes where HCS CellMask Blue and Cell Tracker Blue (Invitrogen) staining were used in addition to DAPI staining, respectively.

**TNT counting.**   A semi-automatized "TNT counting" tool in ICY software was used for TNT counting experiments as previously described [20,129]. Briefly, total number of cells were counted in each image, and the connection between each cell pair was evaluated by scanning through the Z-stacks. Cell pairs that are connected by a protrusion that fits to certain criteria that are used to discriminate TNTs from other cellular protrusions (such as having thin, straight, and uninterrupted connections that are not touching to the substratum) were counted as "TNTs" and connected by a line using a freehand line tool. Total number of cells and number of TNT-connected cells were automatically counted by the tool. Thus, data were presented as percentage of TNT-connected cells.

**Lysosome size (on-section CLEM).**   FM and EM images from the same section were overlaid and correlated with each other. LAMP1 immunogold labeled organelles (lysosomes) were selected blindly, and their size was measured using the "lasso" tool in Fiji. Afterwards, the lysosomes were clustered based on presence of α-syn fibrils, and their size distribution was plotted.

**Lysosome size (SR microscopy).**   Diameter of lysosomes were measured by using freehand straight-line tool of Fiji software. For each condition (control lysosomes and lysosomes containing α-syn or not), SR images were zoomed, and diameter of randomly selected lysosomes was measured.

**Lysosome number.**   The average number of lysosomes in control cells and in cells treated with α-syn fibrils were counted by using the "Spot detector" tool in ICY software.

**Object-based 3D colocalization analysis (Imaris software).**   ROI for each was created by using "Surface" tool in the Surpass mode in Imaris. For each cell, puncta of interest (Gal3-GFP, α-syn-GFP, α-syn fibrils, and LAMP1+ puncta) in different colors were detected by "Spots" tool. Then colocalization was performed first for each pair of interest and then for triple presence of the puncta by using "Colocalization" tool.

## Statistical analysis

Data collected at least from 3 independent experiments were presented as mean ± SEM. Exact number of experiments and number of cells analyzed per condition for each experiment is presented in the figure legends. All data were first subjected to a D'Agostino–Pearson omnibus normality test. Values having Gaussian distribution were further analyzed by 2-tailed $t$ test and 1-way ANOVA followed by Tukey multiple comparison tests for paired and multiple comparisons, respectively. Values having non-Gaussian distribution were analyzed by Whitney–Mann $U$ test and Kruskal–Wallis nonparametric ANOVA test followed by Dunn multiple comparison tests for paired and multiple comparisons, respectively. Paired set of data were analyzed by paired $t$ test.

## Supporting information

**S1 Fig. EM images of control CAD cells. (A)** Resin-embedded EM images of control CAD cells. Three examples of control lysosomes (indicated by blue squares) selected from different regions (indicated by red squares) of 3 different cells are presented. Scale bar for the cells: 5 μm, for the selected regions: 1 μm, and for lysosomes: 0.2 μm. **(B)** On-section EM images of control CAD cells immunogold labeled with LAMP1[10]. Two examples of lysosomes were presented in insets selected from regions indicated by red and blue squares. Scale bar: 1 μm (for the insets: 0.2 μm). CAD, Cath.a-differentiated; EM, electron microscopy; LAMP, lysosome-associated membrane protein.
(TIF)

**S2 Fig. α-Syn fibrils are transferred from donor to acceptor cells inside of lysosomes through TNTs.** Representative image of a TNT having a lysosome containing α-syn fibrils which is formed between α-syn loaded donor cell for 18 hours **(D)** and H2B-GFP transfected acceptor cell **(A)** in 24 hours of CAD-CAD coculture. Z projection of donor and acceptor cells and lysosomes labeled with LAMP1 is presented (upper panels). Z projection of merged image with an additional staining of HCS CellMask Blue, a bottom section (section no: 3) where TNT is not visible and an upper section (section no: 8) where TNT is visible are presented (middle panels). Arrows in the orthogonal view of the upper section and in insets are indicating the lysosome containing α-syn fibrils inside of the TNT (lower panels). Scale bar: 10 μm (for insets: 5 μm). α-syn, α-synuclein; CAD, Cath.a-differentiated; LAMP, lysosome-associated membrane protein; TNT, tunneling nanotube.
(TIF)

**S3 Fig. Schematic presentation of experimental designs. (A)** CAD cells were transiently transfected with Gal3-GFP (1), α-syn-GFP (2), or TFEB-WT-GFP (3) and treated with Alexa 568–tagged α-syn fibrils for 18 hours; cells were then analyzed for the presence of Gal3-GFP puncta, α-syn-GFP puncta, or nuclear TFEB, respectively. **(B)** Donor CAD cells were transiently transfected with LAMP1-GFP and either treated with Alexa 568–tagged α-syn fibrils for 18 hours (coculture prepared from these donors was referred to as "α-syn coculture") or left untreated (coculture prepared from these donors was referred to as "control coculture"). Donor cells were then cocultured with acceptor CAD cells transiently transfected with H2B-mCherry for 24 hours. Efficiency of the LAMP1-GFP+ lysosome transfer was measured in each condition. **(C)** Donor CAD cells were transiently transfected with Arl8b-GFP (coculture prepared from these donors was referred to as "Arl8b coculture") or not (coculture prepared from these donors was referred to as "control coculture") prior to be loaded with Alexa 568–tagged α-syn fibrils for 18 hours. Arl8b-GFP expressing donors (having more peripheral lysosomes) and control donors were then cocultured with acceptor CAD cells that were transiently transfected with H2B-GFP for 24 hours. Efficiency of the α-syn fibrils' transfer was measured in each condition. **(D)** Donor HeLa cells were transiently transfected with Arl8b-GFP (coculture prepared from these donors was referred to as "Arl8b coculture") or not transfected (coculture prepared from these donors was referred to as "control coculture") prior to be loaded with Alexa 568–tagged α-syn fibrils for 18 hours. Arl8b-GFP expressing donors (having more peripheral lysosomes) and control donors were then cocultured with acceptor HeLa cells that were transiently transfected with H2B-GFP for 24 hours. Efficiency of the α-syn fibrils' transfer was measured in each condition. **(E)** Three donor CAD cell populations were prepared: untreated control cells, cells pretreated with sicontrol (scramble), or with siArl8b. Cocultures were referred to as "control," "sicontrol," and "siArl8b" cocultures to distinguish the treatments applied to the donor cells. Donors having perinuclear lysosomes (siArl8b pretreated) and control donors (sicontrol pretreated or untreated) were loaded with Alexa 568–tagged α-syn fibrils for 18 hours prior to be cocultured with acceptor CAD cells transfected with H2B-GFP for 24 hours. Efficiency of the α-syn fibrils' transfer was measured in each condition. **(F)** Donor control cells (WT Hela) or myrlysin-KO HeLa cells or KIF1B-5B dKO HeLa cells were treated with Alexa 568–tagged α-syn fibrils for 18 hours. Control donors (WT HeLa) and donors having perinuclear lysosomes (myrlysin-KO and KIF1B-5B dKO HeLa cells) were then cocultured with the acceptor WT HeLa cells transiently transfected with H2B-GFP for 24 hours. Cocultures were referred to as "control," "myrlysin-KO," and "KIF1B-5B dKO" cocultures to distinguish the treatments applied to the donor cells. Efficiency of the α-syn fibrils' transfer was measured in each condition. **(G)** Donor CAD cells were either transfected with H2B-mCherry (coculture prepared from these donors was referred to as "control

coculture") or treated with Alexa 568–tagged α-syn fibrils for 18 hours (coculture prepared from these donors was referred to as "α-syn coculture") were cocultured with acceptor CAD cells were either transiently transfected with Gal3-GFP (1) or α-syn-GFP (2). Donor and acceptor cells were then cocultured for 24 hours. Acceptors cells were analyzed for the presence of Gal3-GFP or α-syn-GFP puncta, respectively. **(H)** Donor CAD cells transiently transfected with Gal3-GFP and treated with Alexa 568–tagged α-syn fibrils for 18 hours were cocultured with acceptor HeLa cells stably expressing Gal3-Turquoise for 24 hours. Cocultures were either immunolabeled with LAMP1 (mouse) antibody or LAMP1 (human) antibody to detect lysosomes derived from donor and acceptor cells, respectively. **(I)** Secretion test: In parallel to coculture preparations where donor and acceptor cells were mixed in 1:1 ratio, donors and acceptor cells were also plated separately, and medium of acceptor cells was replaced by the conditioned medium collected from donor cells. Acceptor cells were then analyzed for the presence of different cargos (e.g., α-syn fibrils, and lysosomes) in order to evaluate cargo transfer from donor to acceptor cells through secretion. α-syn, α-synuclein; CAD, Cath.a-differentiated; Gal3, Galectin-3; KO, knockout; LAMP, lysosome-associated membrane protein; TFEB, transcription factor EB; WT, wild-type.
(TIF)

**S4 Fig. α-Syn fibrils do not induce lysophagy.** Representative confocal images of CAD cells control, treated with 1 mM LLOMe for 3 hours or treated with Alexa 568–tagged α-syn fibrils for 18 hours and immunolabeled with LC3-Alexa 488, Ubiquitin FK2-Alexa 647, and LAMP1-CF405M (pseudo colored in gray) antibodies. Light blue arrows indicate lysosomes under lysophagy. $n$ = 3 (60 cells analyzed per condition). Scale bar: 10 μm. α-syn, α-synuclein; CAD, Cath.a-differentiated; LAMP, lysosome-associated membrane protein; LC3, Microtubule-associated protein 1A:1B-light chain 3; Ub, Ubiquitin; LLOMe, L-leucyl-L-leucine methyl ester.
(TIF)

**S5 Fig. Time frames of S1 and S2 Movies of CAD cells overexpressing soluble α-syn-GFP treated or not with Alexa 568–tagged α-syn fibrils up to 4 hours. (A)** Time frames at 0/1/2/3/4 hours of the S1 Movie of CAD cells overexpressing soluble α-syn-GFP monitored after the administration of α-syn fibrils. **(B)** Time frames at 0/1/2/3/4 hours of the S2 Movie of CAD cells overexpressing soluble α-syn-GFP. α-syn, α-synuclein; CAD, Cath.a-differentiated.
(TIF)

**S6 Fig. Lysosome positioning affects the efficiency of α-syn fibrils' transfer in HeLa cells. (A)** Representative confocal images of donor and acceptor HeLa cells after 24 hours of coculture. Donor cells were either treated with Alexa 568–tagged α-syn fibrils for 18 hours (referred as "control coculture") or were transfected with Arl8b-GFP prior to be treated with α-syn fibrils (referred as "Arl8b-GFP coculture"). Acceptor cells were transfected with H2B-GFP. Moreover, 24 hours later, cocultures were labeled with HCS CellMask Blue and DAPI. Arrows indicate the α-syn fibrils received by acceptor cells, and asterisks indicate the cells having α-syn fibrils in each coculture condition. Scale bar: 10 μm. **(B)** % of acceptor cells received α-syn fibrils in control coculture (28 ± 3%), Arl8b-GFP coculture (60 ± 5%), control secretion (1 ± 1%), and Arl8b-GFP secretion (8 ± 3%) is presented (left graph); average number of α-syn puncta in control coculture (3 ± 1), Arl8b-GFP coculture (5 ± 1), control secretion (1 ± 0.0), and in Arl8b-GFP secretion (2 ± 0.2) is presented (right graph). Mean ± SEM, $n$ = 3 (70 acceptor cells per condition). ns = not significant, ***$P$ = 0.0001 by Kruskal–Wallis nonparametric ANOVA test followed by Dunn multiple comparison tests. **(C)** Representative confocal images of donor and acceptor HeLa cells after 24 hours of coculture. WT, myrlysin-KO, and KIF1B-5B dKO HeLa cells were loaded with Alexa 568–tagged α-syn fibrils (referred as control,

myrlysin-KO, and KIF1B-5B dKO cocultures, respectively) and cocultured with acceptor cells transfected with H2B-GFP. Cocultures were labeled with HCS CellMask Blue and DAPI. Arrows indicate the α-syn fibrils received by acceptor cells, and asterisks indicate the cells having α-syn fibrils in each coculture condition. Scale bar: 10 μm. **(D)** % of acceptor cells received α-syn fibrils in control coculture (29 ± 2%), myrlysin-KO coculture (16 ± 2%), KIF1B-5B dKO coculture (12 ± 2%), control secretion (5 ± 2%), myrlysin-KO secretion (6 ± 1%), and KIF1B-5B dKO secretion (6 ± 1%) is presented (left graph); average number of α-syn in control coculture (2 ± 0.3), myrlysin-KO coculture (2 ± 0.3), KIF1B-5B dKO coculture (2 ± 0.4), control secretion (1 ± 0.4), myrlysin-KO secretion (1 ± 0.2), and KIF1B-5B dKO secretion (1 ± 0.2) is presented (right graph). Mean ± SEM, $n = 3$ (90 acceptor cells per condition). ns = not significant, $^{**}P = 0.016$, $^{****}P < 0.0001$ by Kruskal–Wallis nonparametric ANOVA test followed by Dunn multiple comparison tests. The data underlying this figure may be found in S1 Data. α-syn, α-synuclein; KO, knockout; WT, wild-type.
(TIF)

**S7 Fig. Validation of CAD-HeLa coculture system with additional information. (A)** Representative confocal image of donor CAD cells loaded with Alexa 568–tagged α-syn fibrils (18 hours) cocultured with HeLa Gal3-Turquoise acceptor cells for 24 hours. In acceptor HeLa cells, arrows indicate α-syn fibrils. Scale bar: 10 μm. **(B)** % of acceptor HeLa cells received α-syn fibrils in coculture (65 ± 3%) and in secretion test (7 ± 2%) is presented (left graph). Mean ± SEM. $^{****}P < 0.0001$ by 2-tailed $t$ test. Average number of α-syn fibrils puncta per acceptor cell in coculture (4 ± 0.2) and in secretion test (2 ± 0.2) is presented (right graph). ns = not significant by Mann–Whitney $U$ test. **(C)** Representative confocal image of donor CAD cells loaded with Alexa 568–tagged α-syn fibrils (18 hours) cocultured with acceptor WT HeLa cells transfected with α-syn-GFP for 24 hours. Red arrows indicate α-syn fibrils, and green arrows indicate α-syn-GFP puncta formation (seeding) in acceptor HeLa cells. Scale bar: 10 μm. **(D)** Representative confocal image of Gal3-GFP transfected and Alexa 568–tagged α-syn fibril loaded donor CAD cells (18 hours) cocultured with HeLa Gal3-Turquoise cells for 24 hours. In donor CAD cell, green arrows indicate Gal3-GFP puncta formation. Scale bar: 10 μm. **(E)** % of donor CAD cells under LMP in control (2 ± 2%) and in α-syn (11 ± 4%) cocultures is presented (left graph); average number of Gal3-GFP puncta in donor cells under LMP in control (10 ± 2) and in α-syn (14 ± 4) cocultures is presented (right graph). Mean ± SEM. ns = not significant, by $^*P = 0.04$ Mann–Whitney $U$ test. **(F)** HeLa Gal3-Turquoise acceptor cell under LMP labeled for LAMP1 human Alexa 647 antibody (pseudo colored in gray). Colocalization between Gal3-Turquoise puncta and LAMP1 (indicated by arrows) is presented in a selected region indicated by the square. Scale bar: 10 μm (for the inset: 1 μm). **(G)** HeLa Gal3-Turquoise acceptor cell under LMP labeled with LAMP1 mouse Alexa 647 antibody (pseudo colored in gray). Colocalization between Gal3-Turquoise puncta and donor's lysosome (white arrows) and colocalization between α-syn fibrils puncta and donor's lysosome (red arrows) are presented in a selected region indicated by a square. Scale bar: 10 μm (for the inset: 1 μm). **(H)** Representative confocal image of Alexa 568–tagged α-syn fibril-treated donor CAD cells for 18 hours cocultured with HeLa Gal3-Turquoise cells (pseudo colored in yellow) for 24 hours. Higher magnification of a selected region (white square) in donor CAD cell is presented. Arrows indicate a Gal3-Turquoise puncta detected in a donor CAD cell. "Merge" image is saturated for the better visualization of the Gal3-Turquoise puncta. Scale bar: 10 μm (for the inset: 2 μm). **(I)** % of donor CAD cells received Gal3-Turquoise puncta in control (0.3 ± 0.3%) and in α-syn (4 ± 1%) cocultures. Mean ± SEM. $^{**}P = 0.003$ by Mann–Whitney $U$ test. In CAD-HeLa coculture model, data were collected from 7 independent experiments in which 4 and 3 of them were labeled with anti-LAMP1 mouse and human antibodies, respectively.

According to the experiment, in average 100 acceptor or donor cells were analyzed for each condition per experiment. The data underlying this figure may be found in S1 Data. α-syn, α-synuclein; CAD, Cath.a-differentiated; Gal3, Galectin-3; LAMP, lysosome-associated membrane protein; LMP, lysosomal membrane permeabilization; WT, wild-type.
(TIF)

**S1 Movie. Seeding of α-syn-GFP aggregates in CAD cells overexpressing soluble α-syn-GFP treated with α-syn fibrils up to 4 hours.** Time frames of the max projection of 36 slides (step size: 0.20 μm) with a total physical thickness of 7.2 μm, with 5 minutes of interval time. Video was acquired using laser 488 in a spinning disk microscope. α-syn, α-synuclein; CAD, Cath.a-differentiated.
(AVI)

**S2 Movie. Monitoring of CAD cells overexpressing soluble α-syn-GFP up to 4 hours.** Time frames of the max projection of 21 slides (step size: 0.48 μm) with a total physical thickness of 9.55 μm, with 15 minutes of interval time. Video was acquired using laser 488 in a spinning disk microscope. α-syn, α-synuclein; CAD, Cath.a-differentiated.
(AVI)

**S1 Data. Numerical raw data.** Numerical data for Figs 1A, 1B, 1D, 2A, 2B, 2C, 2D, 3B, 3C, 4B, 4F, 5B, 5C, 5D, 5E, 5G, 5H, 6A, 6B, 6D, 7B, 7C, 7E, 7F, 8B, 9B, 9G, 10B, 10D,10G and 10I and S6B, S6D, S7B, S7E, and S7I Figs are presented in separate Excel sheets that are combined in a single Excel file.
(XLSX)

**S1 Raw Images. Raw western blot images.** Raw western blot images related to Figs 2A, 6C and 7C are presented.
(PDF)

# Acknowledgments

The authors thank all the lab members for useful discussion. We are grateful to Institut Pasteur Image Analysis Hub (IAH) for their help in quantitative image analysis and Audrey Salles from Unit of Technology and Service Photonic BioImaging platform (UTechS PBI, Institut Pasteur) and Cécile Leduc from Cell Polarity, Migration and Cancer Unit, Institut Pasteur for their help in SIM acquisitions. We thank Alexandre Dufour for writing the script for the lysosomal positioning analysis, Eric Kowarz for the pSBbi-PUR and pCMV (CAT) T7-SB100 plasmids, Andrea Ballabio (TIGEM) for TFEB-WT-GFP and TFEB-mut-GFP plasmids, and Jost Enninga (Institut Pasteur) and Kristine Schauer (Institut Curie) for discussion. We also acknowledge Cilia de Heus (UMCU) for her assistance in CLEM experiments.

# Author Contributions

**Conceptualization:** Aysegul Dilsizoglu Senol, Maura Samarani, Chiara Zurzolo.

**Data curation:** Aysegul Dilsizoglu Senol, Maura Samarani, Sylvie Syan, Nalan Liv, Judith Klumperman.

**Formal analysis:** Aysegul Dilsizoglu Senol, Maura Samarani, Nalan Liv, Judith Klumperman.

**Funding acquisition:** Chiara Zurzolo.

**Methodology:** Patricia Latour-Lambert.

**Resources:** Carlos M. Guardia, Takashi Nonaka, Masato Hasegawa, Juan S. Bonifacino.

**Supervision:** Chiara Zurzolo.

**Writing – original draft:** Aysegul Dilsizoglu Senol, Maura Samarani.

**Writing – review & editing:** Aysegul Dilsizoglu Senol, Maura Samarani, Carlos M. Guardia, Nalan Liv, Juan S. Bonifacino, Chiara Zurzolo.

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
