## [Editor Report · Decision Letter 0]

30 Sep 2020

Dear Dr Zurzolo, 

Thank you for submitting your manuscript entitled "α -Synuclein fibrils affect lysosome structure, function and positioning to propagate misfolding through tunneling nanotubes" for consideration as a Research Article by PLOS Biology.

Your manuscript has now been evaluated by the PLOS Biology editorial staff, as well as by an academic editor with relevant expertise, and I'm writing to let you know that we would like to send your submission out for external peer review.

Please re-submit your manuscript within two working days, i.e. by Oct 02 2020 11:59PM.

Kind regards,

Roli Roberts

Senior Editor

PLOS Biology

---

## [Decision Letter · Decision Letter 1]

17 Nov 2020

Dear Dr Zurzolo,

Thank you very much for submitting your manuscript "α-Synuclein fibrils affect lysosome structure, function and positioning to propagate misfolding through tunneling nanotubes" for consideration as a Research Article at PLOS Biology. Your manuscript has been evaluated by the PLOS Biology editors, an Academic Editor with relevant expertise, and by three independent reviewers.

You will see that all three reviewers are broadly positive about your study, your findings, and the strength of support for your claims. However, they also raise a number of concerns and requests that should be addressed for further consideration. Of reviewer #1's two optional requests, we do not require you to address the first request (about Gaucher's disease) - clearly this would be of interest to those in the field, but we think that it is outside the scope of the current study. However, their second request (about the role of TFEB) is more relevant, overlaps with a query from reviewer #2, and we ask you to address this one. All other concerns raised should be addressed.

In light of the reviews (below), we will not be able to accept the current version of the manuscript, but we would welcome re-submission of a much-revised version that takes into account the reviewers' comments. We cannot make any decision about publication until we have seen the revised manuscript and your response to the reviewers' comments. Your revised manuscript is also likely to be sent for further evaluation by the reviewers.

We expect to receive your revised manuscript within 3 months. 

**IMPORTANT - SUBMITTING YOUR REVISION**

*Re-submission Checklist*

*Published Peer Review*

*PLOS Data Policy*

*Blot and Gel Data Policy*

Sincerely,

Roli Roberts

Senior Editor,

rroberts@plos.org,

PLOS Biology

REVIEWERS' COMMENTS:

Reviewer #1:

[identifies himself as Carmine Settembre]

The work entitled "α-synuclein fibrils affect lysosome structure, function and positioning to propagate misfolding through tunneling nanotubes" by Chiara Zurzolo's group describes in great details the functional relationship between α-synuclein and lysosomes. In particular they have investigated the consequences of α-synuclein accumulation on lysosome functions and how lysosomes participates to aggregate formation and spreading of α-synuclein. This is an important work, which might have implications for the understanding of basic principles behind the pathogenesis of α-synucleinopathies, such as Parkinson disease. The authors performed a detailed characterization of the described pathway through the extensive use of very high quality microscopy approaches. The data are crystal clear. The manuscript is very well written and easy to follow.

In my view the manuscript can be accepted in this current form. However, I would like to suggest two additional experiments that, in my opinion, might further improve the relevance and the mechanistic insights of this elegant study. 

1) There is a known relationship between Parkinson and Gaucher disease, a lysosomal storage disorders caused by mutations in GBA. Notably, heterozygous Gaucher carriers have an increased risk to develop Parkinson. The observations that α-synuclein accumulation impairs lysosomal functions suggest a synergistic effect between a-syn and GBA haploinsufficiency. I think it would be interesting to study whether α-synuclein dependent lysosomal dysfunction and LMP is exacerbated in GBA-KO cells and whether these cells have enhanced spreading.

2) The induction of LMP has been recently shown promote nuclear translocation of TFEB (Nakamura et al. NCB 2020), which is known to influence lysosomal motility and exocytosis (Medina et al. DEV CELL 2015 and many others). Furthermore, TFEB overexpression promotes a-syn clearance (Decressac 2013 PNAS). Thus, I think it might be interesting to check whether the lysosomal loading of a-synuclein triggers TFEB nuclear translocation and whether this participates to lysosomal peripheral distribution upon a-synuclein loading. 

Reviewer #2:

The paper by Senol et al proposes that lysosomes play key roles in the transfer of pathogenic alpha-synuclein fibrils between neuronal cells. Through a combination of super-resolution imaging and electron microscopy, the authors provide evidence that lysosomal membrane permeabilization is key for the intracellular propagation of fibrils taken up from the culture medium. Moreover, they propose that lysosomes loaded with alpha-synuclein fibrils can move between cells via nanotubes connecting donor and acceptor cells. 

Overall, the manuscript proposes an intriguing model for alpha-synuclein propagation that is supported by high-quality experiments. 

Some points require additional clarification and/or experimental support, as detailed below.

1- Fig. 4 shows fibrin-induced formation of a-Syn-GFP puncta, which are furthermore shown to localize near or within lysosomes. This is interpreted as evidence of seeding at lysosomes, however analysis was conducted at a single time point post-treatment (18h), thus it is still possible that a-Syn-GFP puncta may have formed elsewhere in the cytoplasm and subsequently migrated or be transported to lysosomes. A more detailed analysis with multiple time points should be carried out.

2- Given the key role of lysosomes as carriers of a-Syn fibrils, does increasing the number of lysosomes (i.e. via overexpression of the master regulator of lysosomal biogenesis, TFEB transcription factor) increase the efficiency of seeding and/or transfer? 

3- a-Syn is shown to cause LMP, which was previously proposed to trigger an autophagy-based repair pathway known as lysophagy (PMID: 23921551). Does a-Syn fibril uptake result in lysophagy (defined as engulfment of Ubiquitin-positive LAMP1-positive lysosomes inside LC3 structures)? Also, does autophagy promote or antagonize a-Syn induced LMP and the transfer of a-Syn-loaded lysosomes between donor and acceptor cells? 

Reviewer #3:

This revised comprehensive study employed an array of imaging techniques as well as other assays to probe the role of lysosomes in the spreading a-syn pathology from cell to cell. This is an area of great interest to neuroscientists who study neurodegeneration. Some of their findings are that a-syn fibrils affect the morphology, function and distribution of lysosomes within CAD (mouse catecholaminergic) cells, and that lysosomes containing a-syn fibrils can be transferred from a donor cell to a recipient. Moreover, because a-syn permeabilizes the lysosomal membrane, once a-syn fibril+/lysosomes are transferred into a naïve recipient cell, the fibrils apparently can escape the compartment and catalyze fibrillization of endogenous, non-toxic a-syn. The lysosomes are transferred from cell to cell via tunneling nanotubes. Points to address are given below. 

Strengths

An array of imaging techniques show a-syn fibrils in or on lysosomes

Functionality of lysosomes were measured by a pH sensitive marker and cathepsin activity

Morphology of lysosomes evaluated by TEM and other imaging techniques

Movement of lysosomes between donor and recipient cells tracked by following different labels.

Attention paid to statistical evaluation of differences in readouts

Weaknesses / or points needing clarification

1) p. 12, lines 417-425:

"Since our results indicate that α-syn fibrils affect lysosomal pH and function (Fig 2 B and D),

to understand whether there is a correlation between lysosome distribution and their

functionality, we next evaluated the distribution of lysosomes containing α-syn fibrils positive

for LysoTracker DR and Magic Red CathB. We found that α-syn fibril-containing LysoTracker

DR and Magic Red CathB positive lysosomes were more perinuclear (54 ± 2% and 56 ± 1%,

respectively; Fig 5 C), suggesting that peripheral lysosomes containing α-syn fibrils are less

functional. 

These results show that α-syn fibrils affect lysosomes distribution towards the cell periphery.

Furthermore, this pool of peripheral lysosomes is less acidic and has less degradative capacity."

Looking at Fig. 5C, it is hard to understand the sentence, "We found that α-syn fibril-containing LysoTracker DR and Magic Red CathB positive lysosomes were more perinuclear (54 ± 2% and 56 ± 1%, respectively; Fig 5 C), suggesting that peripheral lysosomes containing α-syn fibrils are less functional." 

In the plot, it looks like the perinuclear and peripheral lysosomes have the same functionality.

2) Discussion, p. 30, lines 972-975: 

"However, as we and others demonstrated that cells form more TNTs under stress such as when

they are treated with H202 to increase oxidative stress (107, 108). In line with these findings,

and more specifically, we have previously reported that CAD cells bearing α-syn fibrils exhibit

sustained increase in ROS levels (18). Therefore, α-syn fibrils can indeed trigger stress in the

cells in order to induce TNT formation."

Cancer cells redistribute lysosomes upon stress from the perinuclear region to the periphery (Steffan J Cell Sci 2010, 123: 1151). It seems the same thing is happening here. HeLa cells are cancer cells. Are CAD cells cancer cells? It is not clear to this reviewer what type of cell they are. It says in the methods that they were used undifferentiated, perhaps this implies that these cells are like SH-SY5Y cells, which are cancer cells. 

Questions:

a) Does lysosome trafficking occur between differentiated donor and differentiated acceptor cells?

b) If lysosome redistribution occurs in cancer cells due to stress, then it is quite likely that lysosome redistribution occurs in the CAD cells due to the stress (increase in ROS?), as stated above. It seems reasonable to determine whether an antioxidant like NAC or TEMPO would inhibit lysosome redistribution in CAD cells treated with a-syn fibrils. Pharmacologically inhibiting lysosomal redistribution to the periphery of fibril-treated donor cells should decrease transfer to naive recipient cells. 

c) In the discussion, it would be helpful if the authors talked about the limitations of their model, i.e.,

i) that cancer cells are used. 

ii) if differentiated CAD cells are not optimal to study TNTs (methods section), isn't the probability low that Lewy body pathology spreads from cell to cell via TNT-guided lysosomes in vivo? You do not have to say it the way it is phrased in this sentence. It's just that you should provide the Reader with the limitations of your work.

3). Bar graphs: Fig. 1A; Fig. 2A-D; Fig. 3 B, C; Fig. 4 B,F; Fig. 5 B-E, G,H; Fig. 6 A,C,E,F; and so on.

It would greatly improve the manuscript if the authors showed all data points for each condition rather than just bars with the attached error bar. 

In Fig. 9 D, G and E (left-hand plots), it would help the Reader if the y-axis ranges were ~40 rather than 100.

---

## [Decision Letter · Decision Letter 2]

30 Apr 2021

Dear Dr Zurzolo,

Thank you for submitting your revised Research Article entitled "α-Synuclein fibrils affect lysosome structure, function and positioning to propagate misfolding through tunneling nanotubes" for publication in PLOS Biology. I have now obtained advice from the original reviewers and have discussed their comments with the Academic Editor. 

Based on the reviews, we will probably accept this manuscript for publication, provided you satisfactorily address the following points:

a) We were wondering whether you could adjust the title to make it more straightforward; perhaps something like "α-Synuclein fibrils subvert lysosome structure and function for the propagation of protein misfolding between cells through tunneling nanotubes"?

b) Many thanks for providing all of the underlying data in the supplementary file “data for publication.” Please could you rename it "S1_Data.xlsx" and cite it clearly in all relevant main and supplementary Figure legends? e.g. "The data underlying this Figure may be found in S1 Data."

We expect to receive your revised manuscript within two weeks. 

*Published Peer Review History*

*Early Version*

Sincerely,

Roli Roberts

Senior Editor,

rroberts@plos.org,

PLOS Biology

We require the original, uncropped and minimally adjusted images supporting all blot and gel results reported in an article's figures or Supporting Information files. We will require these files before a manuscript can be accepted so please prepare and upload them now. Please carefully read our guidelines for how to prepare and upload this data: https://journals.plos.org/plosbiology/s/figures#loc-blot-and-gel-reporting-requirements 

DATA NOT SHOWN?

REVIEWERS' COMMENTS:

Reviewer #1:

[identifies himself as Carmine Settembre]

I have no further comments. 

Reviewer #2:

The authors have addressed my previous concerns in full. I recommend immediate acceptance and publication of this interesting and well executed manuscript.

Reviewer #3:

The authors addressed my issues. The manuscript looks good. I have no further issues.

---

## [Editor Report · Decision Letter 3]

13 May 2021

Dear Dr Zurzolo,

On behalf of my colleagues and the Academic Editor, Sharon Tooze, I'm pleased to say that we can in principle offer to publish your Research Article "α -Synuclein fibrils subvert lysosome structure and function for the propagation of protein misfolding between cells through tunneling nanotubes" in PLOS Biology, provided you address any remaining formatting and reporting issues. These will be detailed in an email that will follow this letter and that you will usually receive within 2-3 business days, during which time no action is required from you. Please note that we will not be able to formally accept your manuscript and schedule it for publication until you have made the required changes.

PRESS: We frequently collaborate with press offices. If your institution or institutions have a press office, please notify them about your upcoming paper at this point, to enable them to help maximise its impact. If the press office is planning to promote your findings, we would be grateful if they could coordinate with biologypress@plos.org. If you have not yet opted out of the early version process, we ask that you notify us immediately of any press plans so that we may do so on your behalf.

Thank you again for supporting Open Access publishing. We look forward to publishing your paper in PLOS Biology. 

Sincerely, 

Roli Roberts

Roland G Roberts, PhD 

Senior Editor 

PLOS Biology